# Photoreceptors generate neuronal diversity in their target field through a Hedgehog morphogen gradient in *Drosophila*

Matthew P Bostock, Anadika R Prasad, Alicia Donoghue, Vilaiwan M Fernandes*

Department of Cell and Developmental Biology, University College London, London, United Kingdom

**Abstract** Defining the origin of neuronal diversity is a major challenge in developmental neurobiology. The *Drosophila* visual system is an excellent paradigm to study how cellular diversity is generated. Photoreceptors from the eye disc grow their axons into the optic lobe and secrete Hedgehog (Hh) to induce the lamina, such that for every unit eye there is a corresponding lamina unit made up of post-mitotic precursors stacked into columns. Each differentiated column contains five lamina neuron types (L1-L5), making it the simplest neuropil in the optic lobe, yet how this diversity is generated was unknown. Here, we found that Hh pathway activity is graded along the distal-proximal axis of lamina columns, and further determined that this gradient in pathway activity arises from a gradient of Hh ligand. We manipulated Hh pathway activity cell autonomously in lamina precursors and non-cell autonomously by inactivating the Hh ligand and by knocking it down in photoreceptors. These manipulations showed that different thresholds of activity specify unique cell identities, with more proximal cell types specified in response to progressively lower Hh levels. Thus, our data establish that Hh acts as a morphogen to pattern the lamina. Although this is the first such report during *Drosophila* nervous system development, our work uncovers a remarkable similarity with the vertebrate neural tube, which is patterned by Sonic Hh. Altogether, we show that differentiating neurons can regulate the neuronal diversity of their distant target fields through morphogen gradients.

*For correspondence:
vilaiwan.fernandes@ucl.ac.uk

## Editor's evaluation

This manuscript uncovers a role for a Hh gradient in the differentiation of neuron types in the lamina of the fly eye, a phenomenon reminiscent of its role in the vertebrate nervous systems. It will be of special interest to those who study optic lobe development, but also of more general interest to developmental neurobiology.

## Introduction

Our nervous systems are composed of more unique cell types than any other organ. This diversity is believed to underlie our ability to process sensory input and perform complex specialised tasks. Invertebrates and vertebrates deploy common developmental strategies to diversify neural cell types (*Holguera and Desplan, 2018*). To explore these, we used the complex but tractable visual system of *Drosophila melanogaster*, which is organised into repeated modular circuits that map the visual field.

The *Drosophila* visual system is composed of the compound eyes and optic lobes, which together contain ~200 neuronal cell types (*Fischbach and Dittrich, 1989*; *Kurmangaliyev et al.*,

2020; *Özel et al., 2021*). Each optic lobe is made up of four distinct neuropils, the lamina, medulla, lobula, and lobula plate. With ~100 cell types, the medulla is the most diverse and complex (*Fischbach and Dittrich, 1989*; *Özel et al., 2021*). Despite this, work in recent years has uncovered that the interplay between spatial patterning, temporal patterning, and Notch-mediated binary fate decisions accounts for medulla diversity (*Bertet et al., 2014*; *Erclik et al., 2017*; *Gold and Brand, 2014*; *Konstantinides et al., 2021*; *Li et al., 2013*). In contrast, much less is known about how diversity is generated in the arguably simpler lamina, which contains only five neuronal cell types (L1-L5).

The lamina is the first neuropil to receive input from the photoreceptors and its development is linked to and dependent on photoreceptor development (*Huang et al., 1998*; *Huang and Kunes, 1998*; *Huang and Kunes, 1996*; *Sugie et al., 2010*; *Umetsu et al., 2006*; *Yogev et al., 2010*). Photoreceptors are born progressively in the wake of a wave of differentiation that sweeps across the eye disc (*Figure 1A*; *Roignant and Treisman, 2009*). Two photoreceptor-derived signals, Hedgehog (Hh) and the EGF Spitz drive lamina development (*Fernandes et al., 2017*; *Huang et al., 1998*; *Huang and Kunes, 1998*; *Huang and Kunes, 1996*; *Sugie et al., 2010*; *Umetsu et al., 2006*; *Yogev et al., 2010*). Hh signalling induces neuroepithelial cells to adopt lamina precursor cell (LPC) identity and express the lamina marker Dachshund (Dac). LPCs do not express classic neuroblast markers but are instead a distinct progenitor type, which undergoes terminal divisions in response to Hh signalling (*Apitz and Salecker, 2014*). Finally, Hh signalling also promotes LPC adhesion to photoreceptor axons, which facilitates their assembly into columns (*i.e.* ensembles of stacked LPC cell bodies associated with photoreceptor axon bundles; *Figure 1A and B*; *Huang and Kunes, 1998*; *Huang and Kunes, 1996*; *Sugie et al., 2010*; *Umetsu et al., 2006*). Later in development, photoreceptor-derived EGF is required for LPCs to initiate neuronal differentiation (indicated by expression of the pan-neuronal marker Embryonic lethal abnormal vision [Elav]; *Figure 1A and B*; *Huang et al., 1998*; *Yogev et al., 2010*); however, this communication is indirect and involves a signalling relay through multiple glial populations, which ultimately induce lamina neuron differentiation through MAPK signalling (*Figure 1A*; *Fernandes et al., 2017*; *Prasad et al., 2022*). Lamina neuron identities are invariant and stereotyped according to their position along the distal-proximal axis of columns; LPCs in the distal-most positions of columns will differentiate into L2s, followed by L3s, then L1s, then L4s, and finally L5s are positioned at the proximal end of columns (*Figure 1A and C*; *Pecot et al., 2014*). In addition, we can distinguish lamina neuron types using a combination of markers: Sloppy paired 2 (Slp2) alone labels L2s and L3s, Brain-specific homeobox (Bsh) alone labels L4s, Seven-up (Svp) together with Slp2 label L1s, and Slp2 together with Bsh label L5s (*Figure 1A and C*; *Pecot et al., 2014*). The LPCs positioned between L4s and L5s do not differentiate but instead are fated to die by apoptosis (*Figure 1A*). Given that MAPK signalling drives differentiation in the lamina indiscriminately for all neuronal types (*Fernandes et al., 2017*; *Prasad et al., 2022*), how these neurons acquire their individual identities has remained elusive.

Here, we showed that Hh signalling activity is graded from high to low along the distal-proximal axis of lamina columns. We examined the distribution of Hh ligand in the lamina and detected a protein gradient decreasing from a high point at the distal end of columns, consistent with the observed gradient of pathway activity. We then used genetic manipulations to modulate Hh signalling activity cell autonomously in LPCs or to disrupt Hh ligand directly and showed that different activity thresholds specify LPCs with distinct identities such that progressively decreasing Hh activity results in more and more proximal neuron identities being specified. These data establish that Hh is a morphogen in the lamina. Photoreceptors are the sole source of Hh to the optic lobes at these developmental stages (*Huang and Kunes, 1996*). We showed that Hh derived from photoreceptors patterns lamina cell identities, thus implying that in addition to inducing LPCs, promoting their assembly into columns, and triggering their differentiation through glial signalling relays (*Fernandes et al., 2017*; *Huang et al., 1998*; *Huang and Kunes, 1998*; *Huang and Kunes, 1996*; *Prasad et al., 2022*; *Umetsu et al., 2006*), photoreceptors also determine the neuronal diversity of the lamina. This is the first example of a morphogen patterning neuronal fates in *Drosophila*; it highlights a remarkable similarity with the ventral neural tube of vertebrates where a gradient of the morphogen Sonic Hedgehog (Shh), a Hh homologue, diversifies cell types (*Placzek and Briscoe, 2018*).

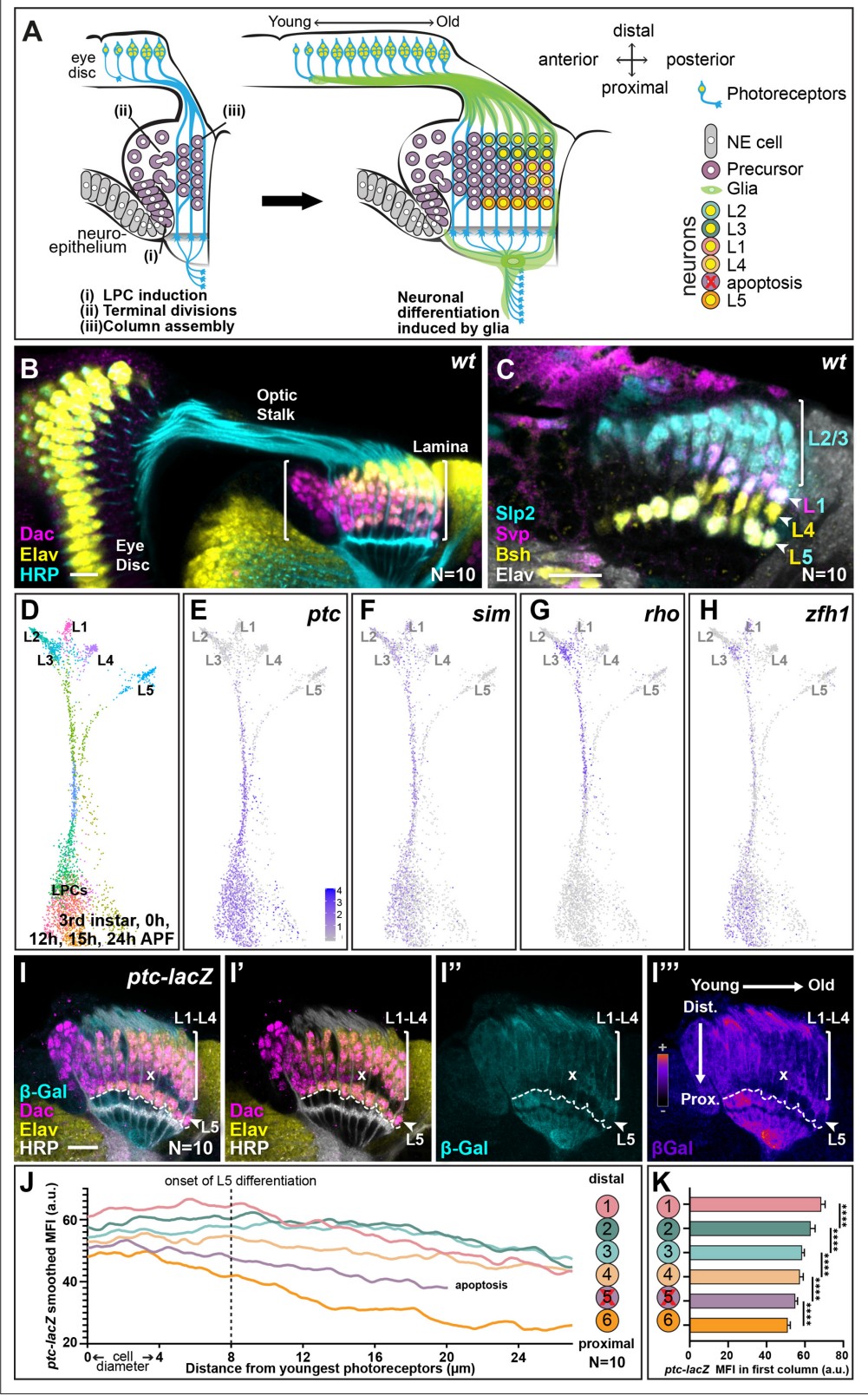

**Figure 1.** Hedgehog (Hh) pathway activity is graded along lamina columns. (**A**) Schematic of the developing lamina. Photoreceptor axons release Hh into the optic lobe, which guides lamina development by driving neuroepithelial cells to develop into lamina precursor cells (LPCs), their terminal divisions, and column assembly. Once assembled into columns of six LPCs each, LPCs differentiate into five distinct neuron types L1–L5 in response

*Figure 1 continued on next page*

*Figure 1 continued*

to MAPK activity that is induced by signals from glia (***Fernandes et al., 2017***; ***Prasad et al., 2022***). Lamina neuron types are positioned stereotypically along the distal to proximal axis of columns: L2, L3, L1, L4, and L5; the LPCs between L4s and L5s undergo apoptosis. (**B**) A wild-type optic lobe and eye disc, with the developing lamina labelled by Dachshund (Dac; magenta), differentiating neurons by Embryonic lethal abnormal vision (Elav; yellow) and photoreceptor axons by Horseradish peroxidase (HRP, cyan). (**C**) A wild-type lamina labelled with lamina neuron-type-specific markers, Sloppy paired 2 (Slp2; cyan), Brain-specific homeobox (Bsh; yellow), and Seven-up (Svp; magenta); all neurons are marked by Elav (white). The Slp2 alone labels L2s and L3s, Bsh alone labels L4s, Svp together with Slp2 label L1s, and Slp2 together with Bsh label L5s. (**D**) Uniform manifold approximation and projection (UMAP) visualisation of LPCs, differentiated L1-L5 neurons, and intermediate stages of differentiation using 150 principal components calculated on the log-normalised integrated gene expression from single-cell RNA sequencing datasets of the third larval instar, 0 hr-, 12 hr-, 15 hr-, and 24 hr-after puparium formation (APF). See ***Figure 1—figure supplement 1A*** for the full integrated dataset. (**E–H**) UMAP visualisation from (D; grey) showing log-normalised expression of Hh signalling targets (blue): (**E**) patched (*ptc*), (**F**), single-minded (*sim*), (**G**) rhomboid (*rho*), and (**H**) zinc finger homeodomain 1 (*zfh1*), which all show higher levels of expression in the convergent stream connecting the LPC cluster with mature L1-L4 neuron clusters rather than the stream connecting the LPC cluster with the L5 neuron cluster. (**I**) An optic lobe expressing *ptc-LacZ* stained for β-galactosidase (β-Gal; cyan), Dac (magenta), Elav (yellow), and HRP (white). The two axes – young to old and proximal to distal are indicated in I''. (**I'''**) shows **I''** in pseudo-colour. The dashed line marks the most proximal surface of the lamina. The 'x' marks the point from which the excess LPCs have been cleared. (**J**) Smoothed (second order with six neighbours; see Materials and methods section) mean fluorescence intensities (MFIs; arbitrary units; a.u.) of *ptc-LacZ* (β-Gal) by distal to proximal cell position as indicated. The L5s and the excess LPCs began with and maintained the lowest levels of β-Gal expression over time. Summary statistics for the raw data in ***Supplementary file 1***. (**K**) Same data as J plotted for the youngest lamina column only. Asterisks indicate significance: p-values <0.0001 from a one-way ANOVA with Dunn's multiple comparisons test. Error bars represent Standard Deviation (SD). Scale bar = 20 μm. See also ***Figure 1—figure supplement 1***.

The online version of this article includes the following figure supplement(s) for figure 1:

**Figure supplement 1.** Identifying and validating additional lamina neuron-type markers.

## Results and discussion

### Hh pathway activity is graded along the distal-proximal axis of LPC columns

For clues into how lamina neuron types are diversified, we integrated published single-cell RNA sequencing (scRNA-seq) datasets at developmental timepoints that span lamina development (***Konstantinides et al., 2021***; ***Kurmangaliyev et al., 2020***; ***Özel et al., 2021***; ***Figure 1—figure supplement 1A***; see Materials and methods section). As published previously, on uniform manifold approximation and projection (UMAP) visualisations L1, L2, L3, and L4 neuronal clusters were closer to each other than they were to the L5 cluster (***Özel et al., 2021***). The LPC cluster was connected to differentiated neurons by two streams of cells: one leading to the L1-L4 clusters and the other leading to the L5 cluster (***Figure 1D***); such convergent streams are thought to represent intermediate states between progenitors and differentiated cells (***Konstantinides et al., 2021***; ***Özel et al., 2021***).

First, we probed these data to identify additional early markers of cell identity for each of the lamina neuron types (beyond Slp2, Bsh, and Svp). Consistent with previous reports, scRNAseq analysis showed that L3 and L4 neurons expressed *earmuff* (*erm*) and *apterous* (*ap*), respectively (***Tan et al., 2015***; ***Ting et al., 2005***). In addition, we found that *reduced ocelli* (*rdo*) was expressed at high levels in L2 neurons and at lower levels and more sporadically in L3 neurons, that L1 neurons expressed *Vesicular glutamate transporter* (*VGlut*) *specifically,* and that L5 and L3 neurons expressed high and low levels of *defective proboscis extension response 8* (*dpr8*), respectively (***Figure 1—figure supplement 1B–F***). We used in situ hybridisation chain reaction (HCR) (***Choi et al., 2018***) to validate marker expression in vivo at 0 hr after puparium formation (0 hr, APF) (***Figure 1—figure supplement 1G–K***), thus providing additional markers for the differentiating lamina neuron types.

Next, we asked which transcripts distinguished L5 precursors from L1 to L4 precursors during development in the scRNA-seq data (see Materials and methods section). Surprisingly, comparing gene expression between the two streams connecting undifferentiated LPCs with L1-L4s and L5s, indicated that several well-established Hh signalling targets were expressed at higher levels in the L1-L4 stream compared with the L5 stream, suggesting that L5 precursors experience lower Hh signalling

than other lamina precursors. These included the direct transcriptional target *patched (ptc),* as well as *single-minded (sim), rhomboid (rho),* and *Zinc finger homeodomain 1 (zfh1)* (*Figure 1E–H*; *Albert et al., 2018*; *Biehs et al., 2010*; *Chen and Struhl, 1996*; *Umetsu et al., 2006*). To validate these data in vivo, we examined the expression of a transcriptional reporter for *ptc (ptc-lacZ)* as a readout of Hh signalling activity (*Chen and Struhl, 1996*; *Tabata and Kornberg, 1994*; *Figure 1I*). We measured the mean fluorescence intensity (MFI) of β-galactosidase (β-Gal) in the lamina at six cell positions corresponding to each LPC position within a column (*i.e.* position 1 corresponded to the most distal cell [prospective L2] and position 6 corresponded to the most proximal cell [prospective L5] in columns; schematic in *Figure 1J*) from young to old columns (anterior to posterior axis). To do so, we generated average intensity projections (from 10 optical slices) obtained from the central lamina and then measured the fluorescence intensity profiles for each of the six cell positions from the youngest column to the oldest column (quantified in *Figure 1J* with summary statistics from a mixed effects linear model in *Supplementary file 1*; see Materials and methods for more detail). We also measured the MFI of β-Gal for each cell along the distal to proximal axis for the youngest column (*Figure 1K*). Consistent with the transcriptomic data where *ptc* expression was lowest in older maturing neurons (*Figure 1E*), *ptc-lacZ* expression eventually decreased along the young to old axis for each of the six (distal-proximal) cell positions (*Figure 1I and J*; N=10). Importantly, in young columns prior to neuronal differentiation, we observed a gradient of *ptc-lacZ* expression, which was highest in the distal lamina and decreased along the distal-proximal axis (*Figure 1K*; one-way ANOVA with Dunn's multiple comparisons test, $p < 0.0001$). Thus, we observed that the direct Hh signalling target, *ptc,* was graded along the distal to proximal length of the youngest column, indicating that there is a gradient of Hh signalling activity in lamina columns prior to neuronal differentiation.

## Hh::GFP is distributed in a protein gradient in the lamina

We sought to determine if graded Hh pathway activity in the lamina arises from a concentration gradient of Hh ligand. We analysed the distribution of Hh protein tagged with superfolder Green Fluorescent Protein (Hh::GFP) expressed under the control of endogenous enhancers of the *hh* locus and inserted into the genome using a bacterial artificial chromosome in a single copy (*Chen et al., 2017*). This transgene was previously reported to rescue a *hh* amorph, indicating that it is a functional *hh* allele (*Chen et al., 2017*). We first used immunohistochemistry (IHC) to detect Hh::GFP and observed higher levels in younger photoreceptor cell bodies compared to older photoreceptor cell bodies in the eye disc, consistent with previous reports (*Figure 2A, A'*; *Huang and Kunes, 1996*). We also observed Hh::GFP in photoreceptor axons in the optic stalk but observed very little signal in photoreceptor axons in the lamina (*Figure 2A, A'*; N=18). Secreted proteins can be particularly sensitive to fixation and washes during IHC, therefore to avoid any confounding artefacts potentially caused by IHC, we visualised Hh::GFP live in cultured brain explants (*Bostock et al., 2020*; *Figure 2B*). We measured Hh::GFP MFI as a function of distal-proximal distance for the youngest lamina column (see Materials and methods section) and observed a gradient polarised from high to low from the distal to proximal ends of columns (*Figure 2B–C*; N=10). These measurements do not distinguish between intracellular and extracellular Hh::GFP, but instead, represent total Hh::GFP. Nonetheless, these data indicate that Hh ligand is present in a distal to proximal gradient in the lamina, consistent with the gradient in Hh pathway activity along lamina columns and suggest that ligand availability is responsible for the activity gradient.

## High and low extremes of Hh pathway activity cell autonomously specify distal and proximal lamina neuron identities, respectively

Hh signalling is known to trigger the early steps of lamina development, including promoting early lamina marker expressions such as *dac* and *sim,* as well as promoting terminal LPC divisions and column assembly (*Huang and Kunes, 1998*; *Huang and Kunes, 1996*; *Sugie et al., 2010*; *Umetsu et al., 2006*). However, no role for Hh in lamina neuron specification or differentiation has been identified thus far. In the vertebrate neural tube, a gradient of Shh signalling specifies ventral progenitor identities (*Placzek and Briscoe, 2018*), therefore, we wondered whether the graded Hh pathway activity we observed could specify neuronal identity in LPCs; in particular, we hypothesised that high Hh signalling levels in LPCs specify distal neuron identity (*i.e.* L2s) and low Hh signalling levels specify proximal neuron identity (*i.e.* L5s).

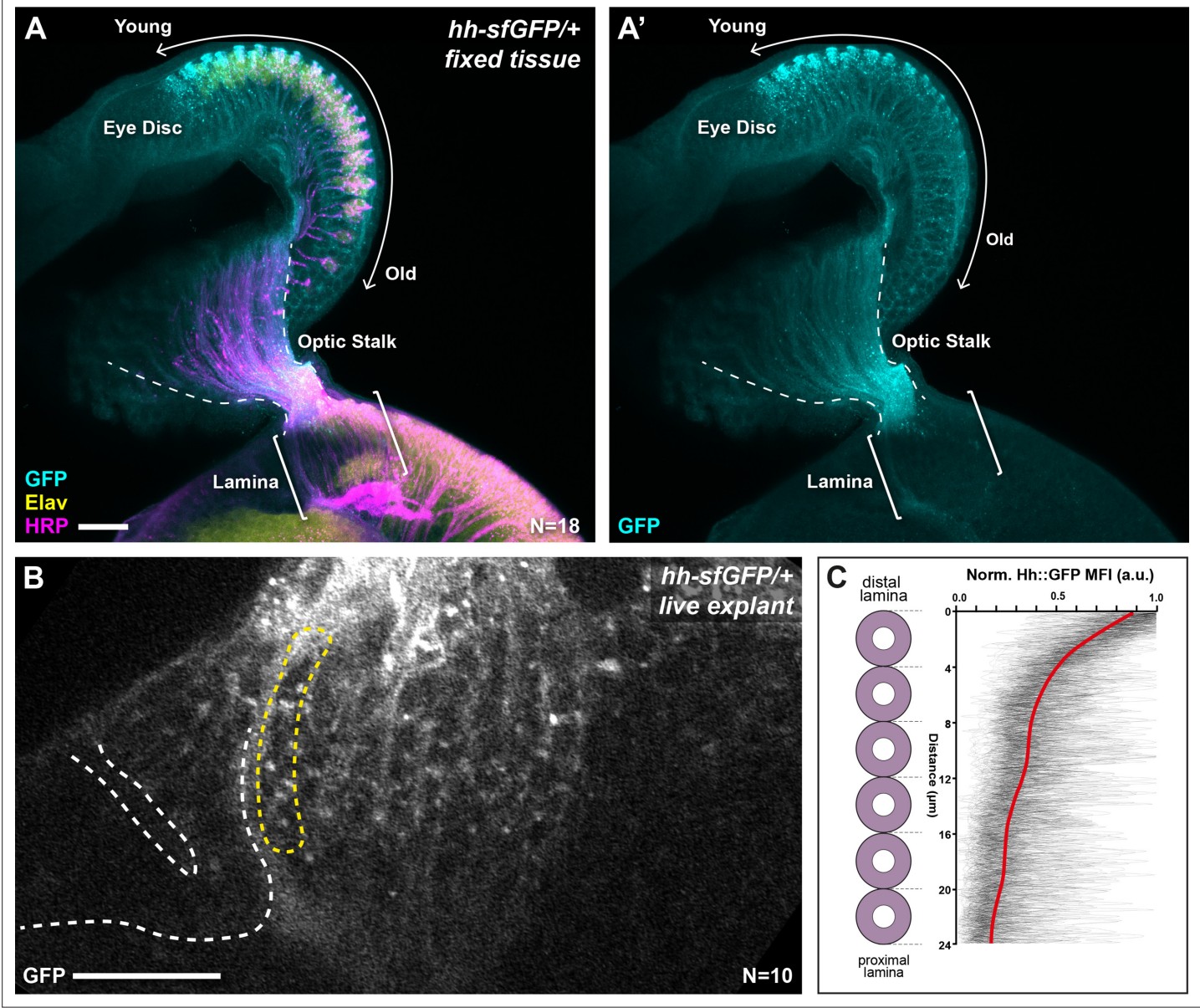

**Figure 2.** Hh::GFP is distributed in a protein gradient in the lamina. (**A**) A maximum intensity projection of a *hh-sfGFP/+* optic lobe and eye disc complex (fixed tissue). Hh::GFP (cyan) detected by immunohistochemistry was present in photoreceptor cell bodies in the eye disc (Embryonic lethal abnormal vision [Elav]; yellow), with higher levels in younger photoreceptors as reported previously (***Huang and Kunes, 1996***) and photoreceptor axons in the optic stalk (Horseradish peroxidase [HRP]; magenta). The Hh::GFP expression decreased rapidly once photoreceptors entered the lamina (brackets). (**B**) A cross-sectional view of the lamina from a live explant of *hh-sfGFP/+*. Hh::GFP puncta were visible more prominently in the distal lamina, with fewer and smaller puncta appearing in proximal regions. White dashed lines mark the lamina furrow. The yellow dashed line marks the youngest column. (**C**) Hh::GFP mean fluorescence intensity (MFI) plots from live explants normalised to the maximum MFI value for each plot (arbitrary units; a.u.) as a function of distance from distal to proximal cell position as indicated for the youngest lamina column (yellow dashed outline in B). The red line shows regression averaging of each of the MFI profiles (see Materials and methods section). Scale bar = 20 µm.

To test this, we asked whether increasing Hh signalling cell autonomously would specify LPCs to differentiate with distal lamina neuron identity (*i.e.* L2). We generated LPCs with a gain of function in Hh pathway activity by inducing positively labelled clones (***Lee and Luo, 2001***) that were homozygous mutants for the negative Hh pathway regulator *ptc* (***Chen and Struhl, 1996***; ***Kalderon, 2005***). The *ptc* mutant clones never contained L5s (Slp2 and Bsh co-expressing cells), L4s (Bsh-only expressing cells), or L1s (Svp and Slp2 co-expressing cells) but instead only contained neurons singly positive for Slp2, indicative of L2/L3 identity (***Figure 3A–B***; 17/17 clones from 15 optic lobes; fully penetrant

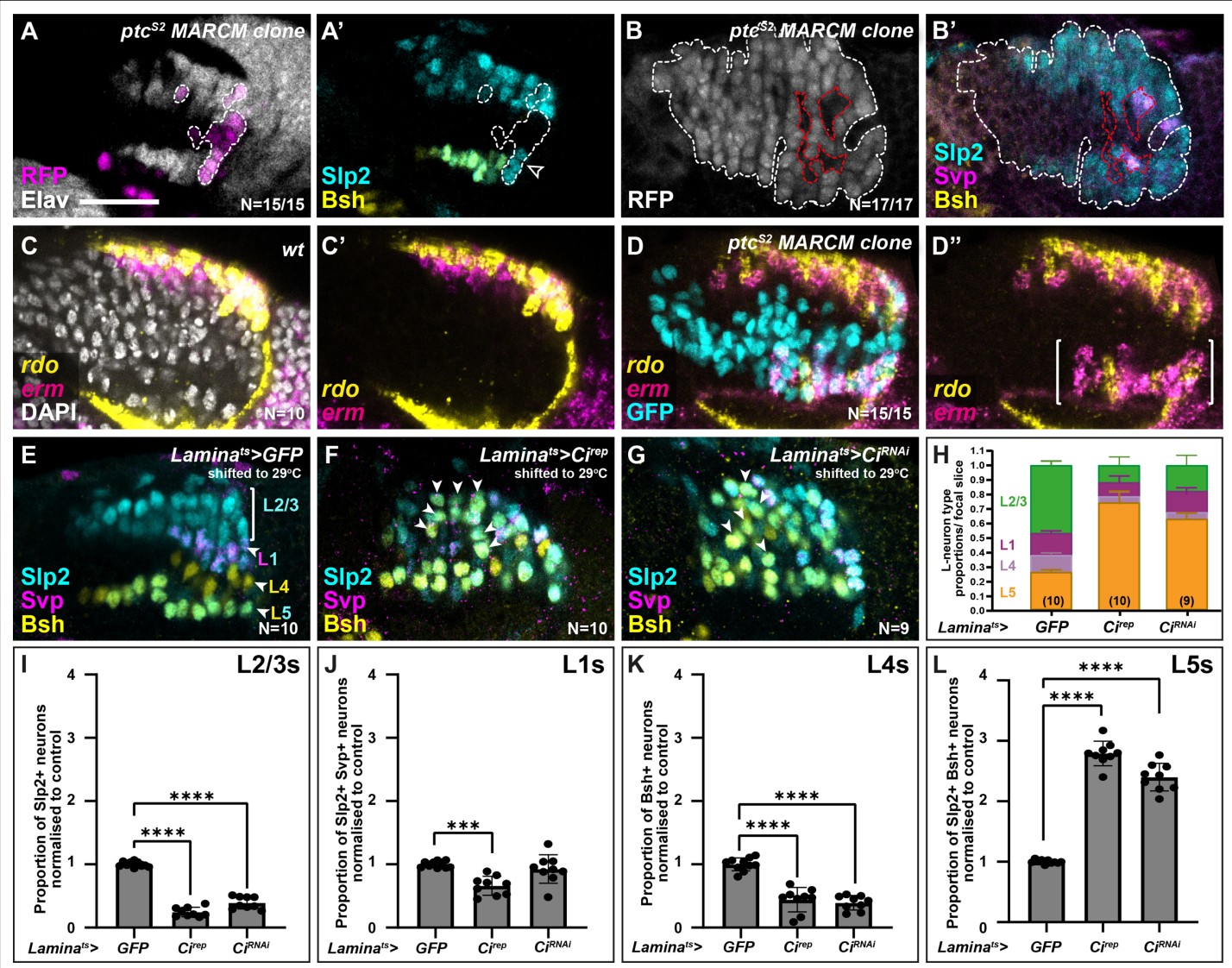

**Figure 3.** High and low extremes of Hedgehog (Hh) pathway activity specify distal and proximal lamina neuron identities, respectively. (**A**) An optic lobe with a small, RFP-positive *ptc^S2* MARCM (mosaic analysis with a repressible cell marker) clone labelled with embryonic lethal abnormal vision (Elav; white), RFP (magenta), sloppy paired 2 (Slp2; cyan), and brain-specific homeobox (Bsh; yellow). Clones in the lamina are outlined by dashed lines. Cells within the clone that were Elav-positive were Slp2-positive but lacked Bsh. (**B**) An optic lobe with a large, RFP-positive *ptc^S2* MARCM clone labelled with RFP (white), Slp2 (cyan), Bsh (yellow), and seven-up (Svp; magenta). Clones in the lamina are outlined by dashed white lines. Cells contained within the lamina that are not part of the clone are outlined by dashed red lines. Cells within the clone that were Elav-positive, were Slp2-positive but not Svp- or Bsh-positive. Note that the Svp-positive cells are not contained within the clone. (**C**) Expression pattern of *reduced ocelli* (*rdo*; yellow) and *earmuff* (*erm*; magenta) in wild-type optic lobes using in situ hybridisation chain reaction. DAPI marks all nuclei (white). (**D**) Expression pattern of *rdo* (yellow) and *erm* (magenta) in optic lobes containing *ptc^S2* mutant clones. Ectopic *rdo* and *erm* were observed in the proximal lamina within the clone. (**E–G**) Lamina-specific misexpression of (**E**) CD8::GFP (control), (**F**) Cubitus interruptus (Ci^rep^), and (**G**) Ci^RNAi^, labelled with lamina neuron-type-specific markers Slp2 (cyan), Bsh (yellow), and Svp (magenta). Ectopic Slp2 and Bsh co-expressing cells (L5s) were recovered in the distal lamina (arrowheads) in (**F and G**). (**H**) Quantifications of the proportion of each lamina neuron type per focal slice aggregated for (**E–F**). (**I–L**) The same as (**H**), normalised to the control and split by lamina neuron type: (**I**) L2-L3s or Slp2-only expressing cells, (**J**) L1s or Slp2 and Svp co-expressing cells, (**K**) L4s or Bsh-only expressing cells, (**L**) L5s or Slp2 and Bsh co-expressing cells (see Materials and methods section). Error bars represent SD. Ns indicated in parentheses in (**H**). One-way ANOVA with Dunn's multiple comparisons test. p***<0.001 and p****<0.0001. Scale bar = 20 µm. See also *Figure 3—figure supplement 1*.

The online version of this article includes the following figure supplement(s) for figure 3:

**Figure supplement 1.** Lamina-specific misexpression of the repressive form of Cubitus interruptus (*Ci^rep^*) and *Ci^RNAi^* perturbs specification.

phenotype). To determine the precise identity of these neurons (L2 or L3), we examined the expression of *rdo* and *erm* by HCR in *ptc* mutant clones in the proximal lamina and found mixed *rdo* and *erm* expression (*Figure 3C–D*; N=15/15 clones), which indicated that both L2 and L3 neurons were present in the clones. Thus, increasing Hh signalling maximally in the lamina resulted in distal neuron specification (mixed L2 and L3) at the expense of L1, L4, and L5 specifications.

Next, we sought to disrupt Hh signalling cell autonomously in the lamina and assess the effects on lamina neuron specification. When we blocked Hh pathway activity completely in clones mutant for *smoothened (smo)*, we did not recover any Dac-expressing LPCs (*Figure 3—figure supplement 1A*; N=15/15 clones; fully penetrant phenotype), consistent with previous reports (*Huang and Kunes, 1998*). Therefore, to reduce Hh pathway activity partially, we used a lamina-specific driver (*R27G05-Gal4*), which drives expression in Dac-expressing LPCs exiting the posterior lamina furrow (*Figure 3—figure supplement 1B*), to express a repressor form of the transcriptional effector of Hh signalling, Cubitus interruptus (Ci; referred to as Ci$^{rep}$) (*Busson and Pret, 2007*). We used a temperature-sensitive Gal80 (Gal80$^{ts}$) to restrict Gal4 activity temporally by raising crosses at the permissive temperature (18°C) for Gal80$^{ts}$ and shifting to the restrictive temperature (29°C) to allow Gal4 activity from the third larval instar (see Materials and methods section). Expressing Ci$^{rep}$ in LPCs led to a 2.8-fold increase in the proportion of Bsh and Slp2 double-positive L5s relative to controls, while the proportion of other lamina neuron subtypes was reduced relative to controls (*Figure 3E, F and H–L* and *Figure 3—figure supplement 1C–F*; one-way ANOVA with Dunn's multiple comparisons test; p-values <0.001; N=10). Moreover, whereas in control laminas L5s were only observed as a single row in the proximal lamina, expressing Ci$^{rep}$ caused Bsh and Slp2 double-positive L5s to be distributed along the entire distal-proximal lengths of columns (*Figure 3F*, N=10). Similarly, knocking down Ci in the lamina by RNA interference (Ci$^{RNAi}$) (*Figure 3G*; N=9) resulted in ectopic L5s (2.4-fold increase relative to controls, one-way ANOVA with Dunn's multiple comparisons test; p<0.0001) and a reduction in the proportion of L2-L3s and L4 lamina neurons relative to controls (*Figure 3H–L*, one-way ANOVA with Dunn's multiple comparisons test; p$^{L2-L3}$<0.0001 and p$^{L4}$<0.0001; see *Figure 3—figure supplement 1C–F* for raw values). Thus, decreasing Hh signalling autonomously within LPCs was sufficient to induce the most proximal cell identity (L5s), throughout the distal-proximal lengths of columns and at the expense of other neuronal types. In summary, disrupting Hh signalling activity cell autonomously to either high or low extremes specified LPCs with either the most distal or proximal lamina neuron identities, respectively.

## Intermediate levels of Hh signalling activity specify intermediate lamina neuron identities

The data presented thus far are consistent with Hh acting as a morphogen to specify cell identities in the lamina, that is, in a concentration-dependent manner; however, our cell-autonomous manipulations only determined the impact of experiencing either the high or low extremes of pathway activity on lamina neuron identity specification. Therefore, we attempted to tune Hh signalling to intermediate levels by using the Gal4/UAS system to express Ci$^{rep}$ in the lamina as before under two temperature conditions (in the presence of Gal80$^{ts}$): 29°C for a strong inhibition of Hh signalling (as above) and 25°C for a milder inhibition. We then evaluated the distribution of lamina neuron types using *rdo*, *erm*, *VGlut*, *ap*, *dpr8*, and *bsh*. Strong inhibition of Hh signalling (*lamina$^{ts}$ >Ci$^{rep}$* at 29°C) led to a 2.2-fold increase relative to controls in the proportion of the lamina that co-expressed *dpr8* and *bsh* (*i.e.* L5s) at the expense of other markers (*Figure 4A–F and J–N*; N$^{dpr8+bsh}$=8; one-way ANOVA with Dunn's multiple comparisons test; p$^{dpr8+bsh}$<0.0001). As before, *dpr8* and *bsh* co-expressing cells (L5s) were no longer restricted to a single row in the proximal lamina but were distributed along the entire distal-proximal lengths of columns (*Figure 4C and F*). In contrast, milder inhibition of Hh signalling (*lamina$^{ts}$ >Ci$^{rep}$* at 25°C) led to a 2.5-fold increase relative to controls in the proportion of the lamina that expressed *VGlut* (*i.e.* L1s) and a 1.9-fold increase relative to controls in the proportion of the lamina that expressed *ap* (*i.e.* L4s), at the expense of *rdo* expressing (*erm*-negative) L2s (*Figure 4A–C– and J–N*; one-way ANOVA with Dunn's multiple comparisons test; p$^{VGlut}$<0.0001; p$^{ap}$<0.01). We obtained similar results when we evaluated the distribution of neuron types with Slp2, Svp, and Bsh (*Figure 4—figure supplement 1*). Thus, intermediate neuron identities (L1s and L4s) were favoured when we tuned Hh signalling to intermediate levels.

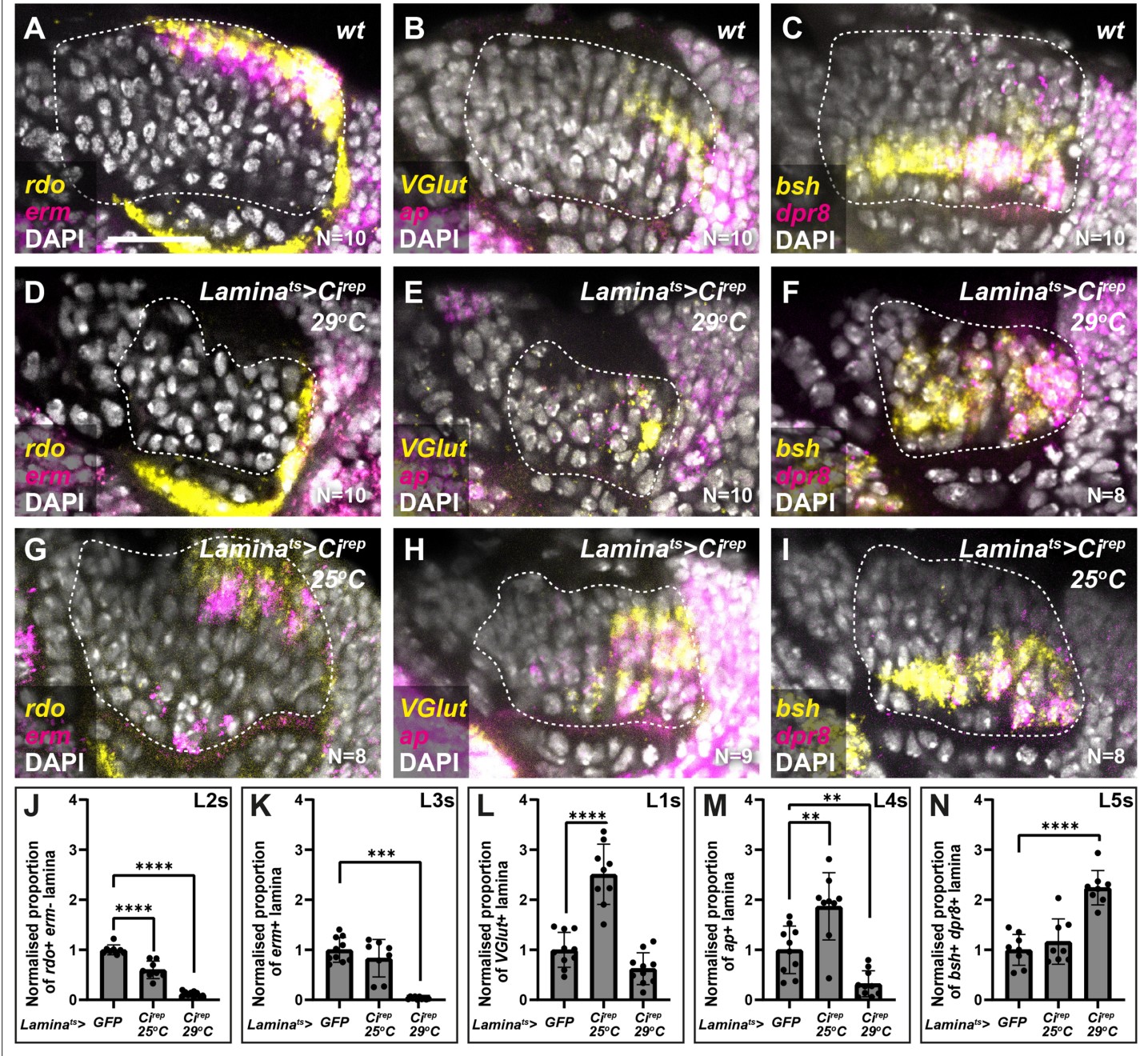

**Figure 4.** Intermediate levels of Hedgehog (Hh) signalling activity specify intermediate lamina neuron identities. (**A–C**) Wild-type optic lobes showing the expression of (**A**) *reduced ocelli* (*rdo*; yellow; L2 and L3s) and *earmuff* (*erm*; magenta; L3s), (**B**) *Vesicular glutamate transporter* (*VGlut*; yellow; L1s) and *apterous* (*ap* magenta; L4s), and (**C**) *brain-specific homeobox* (*bsh*; yellow) and *defective proboscis extension response 8* (*dpr8*; magenta) (co-expressing cells are L5s) by hybridisation chain reaction. DAPI marks all nuclei in white. (**D–F**) Optic lobes from *Lamina^ts^ >Ci^irep^* at 29°C (*i.e.* strong Hh signalling inhibition) labelled for (**D**) *rdo* (yellow) and *erm* (magenta), (**E**) *VGlut* (yellow) and *ap* (magenta), and (**F**) *bsh* (yellow) and *dpr8* (magenta). DAPI in white. (**G–I**) Optic lobes from *lamina^ts^ >Ci^irep^* at 25°C (*i.e.* milder Hh signalling inhibition) labelled for (**G**) *rdo* (yellow) and *erm* (magenta), (**H**) *VGlut* (yellow) and *ap* (magenta), and (**I**) *bsh* (yellow) and *dpr8* (magenta). DAPI in white. (**J–N**) Quantifications of (**A–I**) represented as the relative area of the lamina expressing (**J**) *rdo,* (**K**) *erm*, (**L**) *VGlut,* (**M**) *ap*, and (**N**) *bsh* and *dpr8* per focal slice, normalised to the control (see Materials and methods section). One-way ANOVA with Dunn's multiple comparison test. $p^{**} < 0.01$; $p^{***} < 0.001$; and $p^{****} < 0.0001$. Error bars represent SD. Scale bar = 20 μm.

The online version of this article includes the following figure supplement(s) for figure 4:

**Figure supplement 1.** Intermediate levels of Hedgehog (Hh) signalling activity specify intermediate lamina neuron identities when evaluated by alternative neuron-type markers.

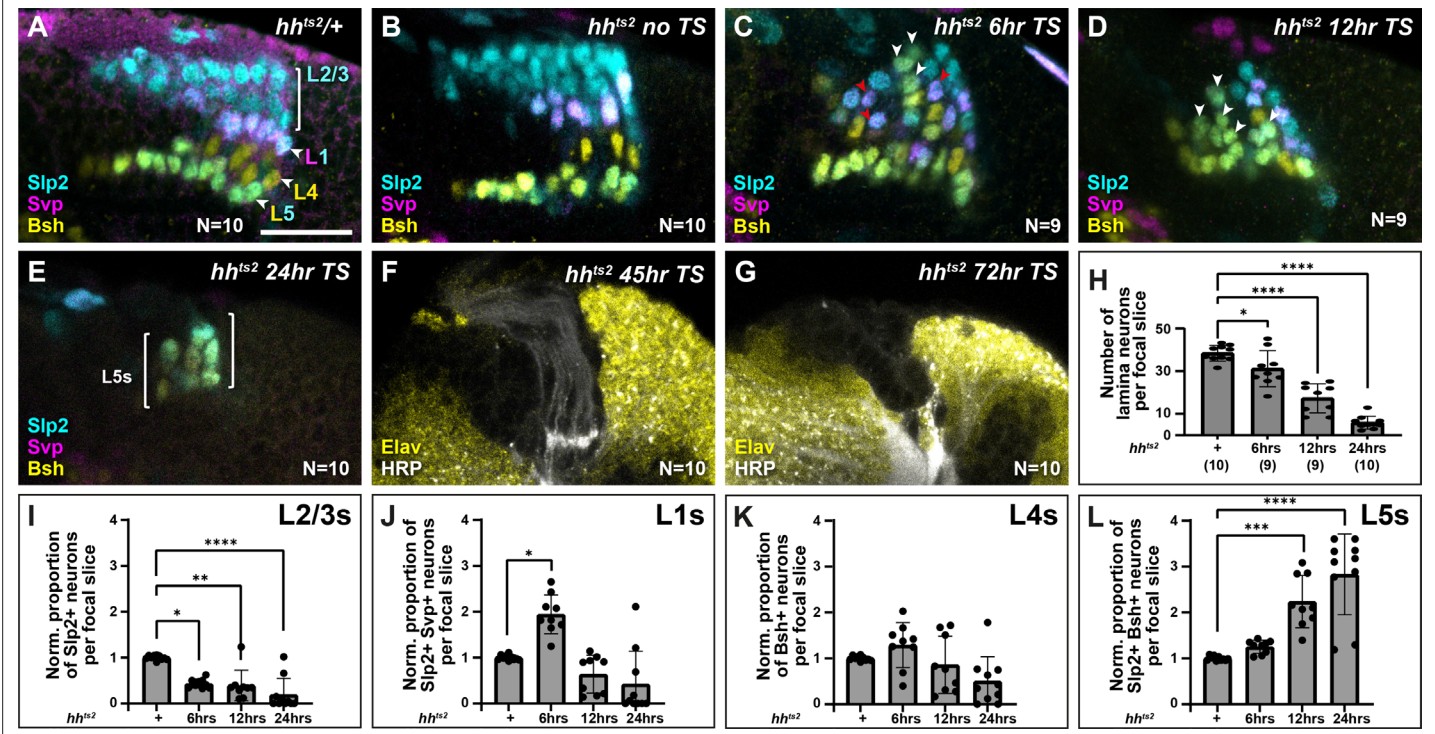

**Figure 5.** Titrating functional Hedgehog (Hh) availability favours distinct neuronal types at different thresholds. Laminas labelled with neuron-type-specific markers Sloppy paired 2 (Slp2; cyan), Seven-up (Svp; magenta), and Brain-specific homeobox (Bsh; yellow) in (A) $hh^{ts2}$/+shifted from the permissive temperature (18°C) to the restrictive temperature (29°C) for 24 hr, (B) $hh^{ts2}$ raised at the permissive temperature (no temperature shift [no TS]), (C–G) $hh^{ts2}$ shifted from the permissive temperature to the restrictive temperature for (C) 6 hr, (D) 12 hr, and (E) 24 hr. The pattern of neuronal differentiation worsened progressively with longer TSs, with fewer neurons differentiating overall. Slp2- and Bsh-positive cells (L5s) were observed in the distal lamina (white arrowheads) for 6-hr, 12-hr, and 24-hr TSs, till most cells present differentiated into L5 neurons for the 24-hr TS. Whereas Slp2 and Svp co-expressing cells (L1) increased only for the 6-hr TS and were distributed throughout the distal-proximal axis (red arrowheads). (F and G) $hh^{ts2}$ shifted from the permissive temperature to the restrictive temperature for (F) 45-hr and (G) 72-hr stained for embryonic lethal abnormal vision (Elav; yellow) and horseradish peroxidase (HRP; white). A few photoreceptor bundles are present but no lamina precursor cells (LPCs) formed under the (F) 45-hr TS condition, whereas neither photoreceptors nor LPCs were present for the (G) 72-hr TS condition. (H) Quantification of the total number of lamina neurons (Elav-positive cells) per focal slice in $hh^{ts2}$/+ and $hh^{ts2}$ shifted from the permissive temperature to the restrictive temperature as indicated. Error bars represent SD. Ns indicated in parentheses. (I–L) Quantifications of the proportions of each lamina neuron type per focal slice and normalised to the control for each $hh^{ts2}$ TS condition. One-way ANOVA with Dunn's multiple comparison test. $p^* < 0.05$; $p^{**} < 0.01$; $p^{***} < 0.001$; and $p^{****} < 0.0001$. Error bars represent SD. Scale bar = 20 μm. *Figure 5—figure supplement 1*.

The online version of this article includes the following figure supplement(s) for figure 5:

**Figure supplement 1.** Fewer photoreceptors and lamina neurons develop when functional Hedgehog (Hh) availability is titrated.

To further test whether different thresholds of Hh ligand determine lamina neuron identities, we sought to manipulate Hh directly and to differing degrees of severity. We predicted that, if Hh was acting as a morphogen, progressively reducing Hh levels should result in progressively more proximal identities. To do so, we used animals harbouring a temperature-sensitive loss of function allele of *hh* ($hh^{ts2}$), which we raised at the permissive temperature (18°C) and then shifted to the restrictive temperature (29°C) for either 6 hr, 12 hr, 24 hr, 45 hr, or 72 hr before dissecting at the white prepupal stage (*Figure 5A–G*, *Figure 5—figure supplement 1A-J*). Consistent with the known role of Hh signalling in photoreceptor and early lamina development (*Huang and Kunes, 1996*), we recovered fewer and fewer photoreceptors and lamina neurons with increasing lengths of temperature shifts (*Figure 5—figure supplement 1A-J*), with no LPCs or lamina neurons recovered for the 45-hr and 72-hr temperature shifts (*Figure 5F and G*; *Huang and Kunes, 1996*). These results indicated that our perturbations effectively titrated the availability of functional Hh with longer periods at the restrictive temperature. Next, we used lamina neuron-type-specific markers to evaluate neuronal cell types under these conditions. We found that the proportion of L2-L3s (Slp2 single-positive neurons) decreased as the availability of functional Hh was titrated to lower and lower levels, whereas the proportion of

L5s (Slp2 and Bsh double-positive neurons) increased (*Figure 5A–E*; quantified in *Figure 5I–L*; one-way ANOVA with Dunn's multiple comparisons test; p-values <0.05). Importantly, the proportion of L1s (Slp2 and Svp double-positive neurons) increased only for the 6-hr temperature shift. Thus, L1 neurons, which normally occupy intermediate lamina column positions were favoured at the expense of distal neuron types at intermediate thresholds of Hh pathway activity (*Figure 5C and J*; one-way ANOVA with Dunn's multiple comparisons test; p-values <0.05). Notably, under these conditions, L1s and L5s were no longer confined to their normal positions within lamina columns but were observed throughout (*Figure 5A–D*). Altogether our data establish that Hh acts as a morphogen to diversify lamina neuron types.

## Photoreceptor-derived Hh specifies lamina neuron identities

Others have reported previously that photoreceptors are the sole source of Hh to the optic lobes during lamina development (*Huang and Kunes, 1996*). We corroborated these results using an enhancer trap in the *hh* locus (*hh-lacZ; hh^{P30}*), which showed that β-Gal expression was restricted to photoreceptors in the eye disc and absent from the optic lobes, except for a few sparse cells near the central brain (*Figure 6A*). We reasoned that if photoreceptor-derived Hh indeed patterns lamina neuron identities, then disrupting Hh expression specifically in photoreceptors should in turn result in lamina neuron patterning defects. Hh secreted by developing photoreceptors is required to drive morphogenetic furrow progression and continued photoreceptor development in the eye disc (*Greenwood and Struhl, 1999*; *Treisman, 2013*). Consequently, removing *hh* expression completely in the eye disc has been shown to block both photoreceptor development (*Greenwood and Struhl, 1999*; *Treisman, 2013*) and lamina development, due to its early role in inducing LPCs (*Huang and Kunes, 1998*; *Huang and Kunes, 1996*). Therefore, we used *GMR-Gal4*, which is expressed after photoreceptor birth together with a previously validated RNAi line (*Kim et al., 2014*; *Ni et al., 2011*; *Sahai-Hernandez and Nystul, 2013*) to partially knock down Hh in photoreceptors, thus bypassing its early roles. Under these conditions, lamina size was not noticeably affected, indicative of a mild knockdown (*Figure 6B and C*). When we evaluated lamina neuron-type identities, we found that Slp2 and Svp expressing cells (L1s) increased by 1.2-fold relative to controls (*Figure 6C*; Mann-Whitney U test; p<0.001; N=10), consistent with more LPCs experiencing intermediate Hh signalling activity. Importantly, L1 neurons were no longer restricted to a single row at intermediate positions in columns but instead were recovered in more distal positions (*Figure 6C*). Thus, partially knocking down Hh expression in photoreceptors disrupted lamina neuron patterning in favour of more intermediate cell identities (L1s), consistent with mildly decreasing Hh signalling to more intermediate levels in the lamina.

Here, we uncovered a previously unknown role for Hh as a morphogen that specifies lamina neuron identities, thus revealing yet another layer of dependence of lamina development on photoreceptor-derived signals. Intriguingly, a mix of L2 and L3 identities were specified when Hh signalling was activated to maximum physiological levels (*Figure 3D*). This suggests that in addition to high Hh signalling levels, additional signals are required to further differentiate L2 from L3 identities. This is the first report of a morphogen patterning neuronal fates in *Drosophila*. It highlights the remarkable evolutionary conservation of this developmental strategy, which is widespread in vertebrates (*Placzek and Briscoe, 2018*; *Sagner and Briscoe, 2019*).

Nonetheless, it is not obvious how the Hh gradient is established in the lamina, such that it is polarised from high to low along the distal-proximal lengths of lamina columns, as photoreceptor axons themselves span this full distance. Axonal trafficking of Hh was shown to be regulated by an evolutionarily conserved motif in its cleaved C-terminal domain (*Chu et al., 2006*; *Daniele et al., 2017*), disruption of which affected neuronal diversity in the lamina specifically (*Chu et al., 2006*). This indicates that axonal trafficking may play a role in establishing the Hh gradient; however, more work is required to uncover the exact molecular mechanisms involved. It is unexpected that neurons, here photoreceptors, define the neuronal diversity of their target field, and raises the possibility that neurons in other contexts could also be patterning their target fields during development through morphogen gradients. Indeed, many vertebrate neuronal types express Shh (*Dakubo et al., 2003*; *Farmer et al., 2016*; *Garcia et al., 2010*; *Gonzalez-Reyes et al., 2012*; *Harwell et al., 2012*; *Petralia et al., 2011*; *Wallace, 1999*); however, their ability to generate instructive gradients to pattern distant neural fields has not been determined thus far.

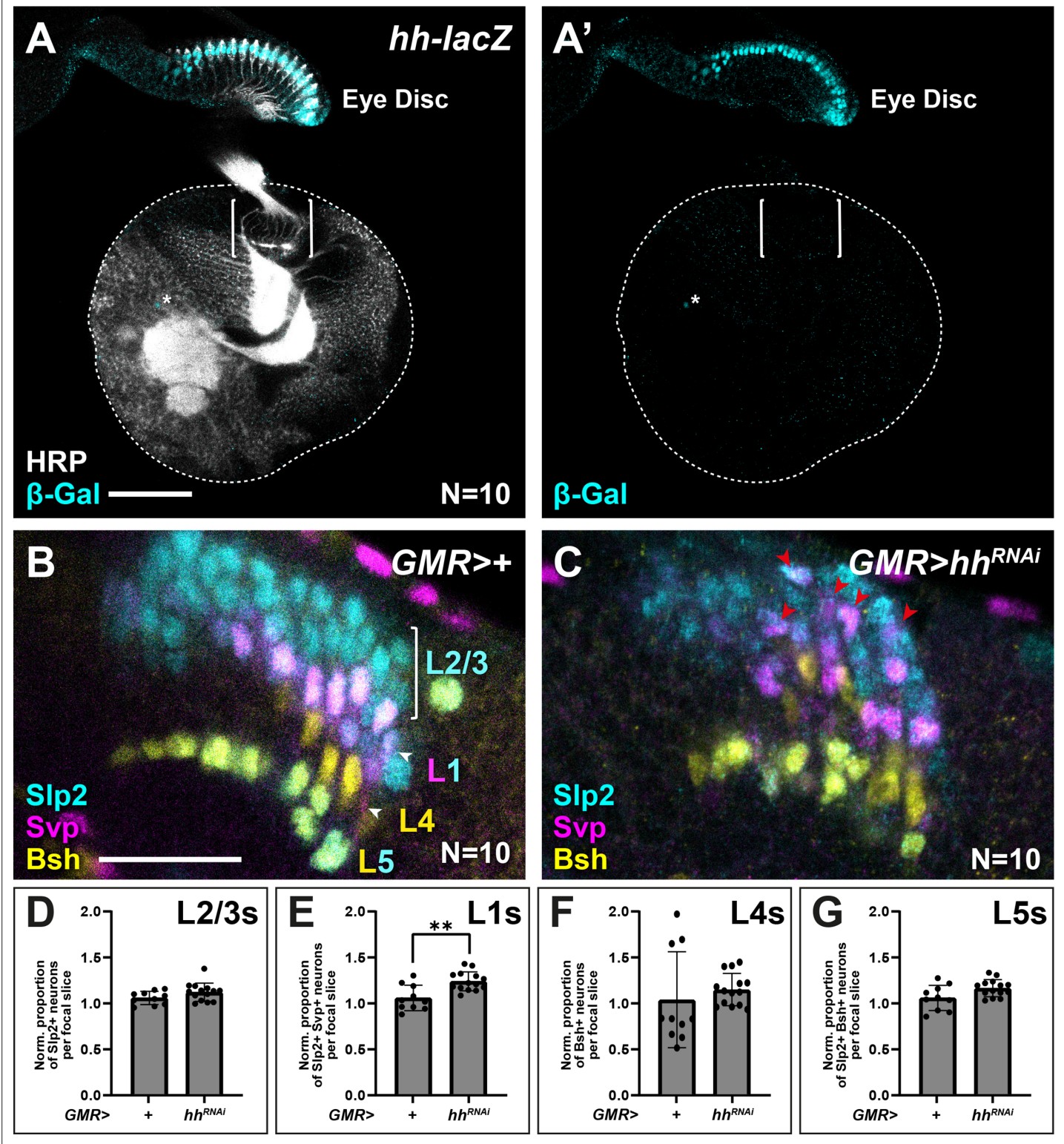

**Figure 6.** Appropriate lamina neuron patterning requires photoreceptor-derived Hedgehog (Hh). (**A**) Expression pattern of *hh-lacZ*. Only photoreceptors in the eye-disc expressed nuclear β-galactosidase (β-Gal; cyan) along with a few sparse cells near the central brain (asterisk). Horseradish peroxidase (HRP) in white. Brackets mark the lamina. (**B–C**) Optic lobes from (**B**) *GMR>+* (control) and (**C**) *GMR>hh^{RNAi}* labelled with lamina neuron-type-specific markers sloppy paired 2 (Slp2; cyan), brain-specific homeobox (Bsh; yellow), and seven-up (Svp; magenta). (**D–G**) Quantifications of the proportion of each lamina neuron type relative to controls per focal slice for (**D**) Slp2-only expressing L2-L3s, (**E**) Slp2 and Svp co-expressing L1s, (**F**) Bsh-only expressing L4s, and (**G**) Bsh and Slp2 co-expressing L5s. One-way ANOVA with Dunn's multiple comparison test. p*<0.05. Error bars represent SD. Scale bar = 20 μm.

# Materials and methods

**Key resources table**

| Reagent type (species) or resource | Designation | Source or reference | Identifiers | Additional information |
|---|---|---|---|---|
| Genetic reagent (*Drosophila melanogaster*) | *Canton S* | Bloomington *Drosophila* Stock Center | BSDC: 64349 | |
| Genetic reagent (*D. melanogaster*) | *ptc-lacZ* | PMID:10769240 | | Gift from D Kalderon |
| Genetic reagent (*D. melanogaster*) | *hh-sfGFP* | Bloomington *Drosophila* Stock Center | BDSC: 86271 | |
| Genetic reagent (*D. melanogaster*) | *FRT42D, ptc$^{S2}$* | Bloomington *Drosophila* Stock Center | BDSC: 6332 | |
| Genetic reagent (*D. melanogaster*) | *ey-Gal80* | Bloomington *Drosophila* Stock Center | BDSC: 35822 | |
| Genetic reagent (*D. melanogaster*) | *Gal80$^{ts}$* | Bloomington *Drosophila* Stock Center | BDSC: 7108 | |
| Genetic reagent (*D. melanogaster*) | *R27G05-Gal4* | Bloomington *Drosophila* Stock Center | BDSC: 48073 | |
| Genetic reagent (*D. melanogaster*) | *UAS-CD8::GFP* | Bloomington *Drosophila* Stock Center | BDSC: 32187 | |
| Genetic reagent (*D. melanogaster*) | *UAS-Ci$^{76}$* | PMID:9215627 | UAS-Ci$^{rep}$ | Gift from D J Treisman |
| Genetic reagent (*D. melanogaster*) | *UAS-Ci$^{RNAi}$* | Bloomington *Drosophila* Stock Center | BDSC: 64928 | |
| Genetic reagent (*D. melanogaster*) | *smo$^3$, FRT40A* | PMID:9811578 | | Gift from D C Boekel |
| Genetic reagent (*D. melanogaster*) | *hh$^{ts2}$* | Bloomington *Drosophila* Stock Center | BDSC: 1684 | |
| Genetic reagent (*D. melanogaster*) | *hh$^{P30}$* | Bloomington *Drosophila* Stock Center | BDSC: 5530 | |
| Genetic reagent (*D. melanogaster*) | *GMR-Gal4* | Bloomington *Drosophila* Stock Center | BDSC: 1104 | |
| Genetic reagent (*D. melanogaster*) | *UAS-hh$^{RNAi}$* | Bloomington *Drosophila* Stock Center | BDSC: 32489 | |
| Genetic reagent (*D. melanogaster*) | *ywhsflp122, tub-gal4, UAS-GFP; FRT40A, Gal80/Cyo* | PMID:10197526 | BDSC: 5192 | Gift from F Schweisguth |
| Genetic reagent (*D. melanogaster*) | *ywhsflp122, tub-gal4, UAS-RFP; FRT42D, Gal80/Cyo* | PMID:10197526 | BDSC: 9917 | Gift from G Struhl |
| Antibody | anti-Dac2-3 (mouse monoclonal) | Developmental Studies Hybridoma Bank | mAbdac2-3 | 1:20 |
| Antibody | anti-Elav (rat monoclonal) | Developmental Studies Hybridoma Bank | 7E8A10 | 1:100 |
| Antibody | anti-Svp (mouse monoclonal) | Developmental Studies Hybridoma Bank | 6F7 | 1:50 |
| Antibody | anti-Slp2 (guinea pig polyclonal) | PMID:23783517 | C Desplan | 1:100 |
| Antibody | anti-Bsh (rabbit polyclonal) | PMID:33149298 | C Desplan | 1:500 |
| Antibody | anti β-galactosidase (mouse monoclonal) | Promega | #Z3781 | 1:500 |
| Antibody | anti-GFP (chicken polyclonal) | EMD Millipore | GFP-1010 | 1:400 |

*Continued on next page*

*Continued*

| Reagent type (species) or resource | Designation | Source or reference | Identifiers | Additional information |
|---|---|---|---|---|
| Antibody | anti-Sim (mouse monoclonal) | PMID:16439478 | T Tabata | 1:20; Originally from DSHB, but no longer produced |
| Antibody | anti-RFP (chicken polyclonal) | Rockland | #600-901-379s | 1:500 |
| Antibody | anti-GFP (rabbit polyclonal) | Thermofisher | #A6455 | 1:500 |
| Antibody | AlexaFluor405-conjugated Anti-HRP (goat polyclonal) | Jackson Immunolabs | 123-475-021 | 1:200 |
| Antibody | AlexaFluorCy3- conjugated anti-HRP (goat polyclonal) | Jackson Immunolabs | 11 23-165-021 | 1:200 |
| Antibody | AlexaFluor647- conjugated anti-HRP (goat polyclonal) | Jackson Immunolabs | 123-605-021 | 1:200 |
| Antibody | Anti-rabbit Alexa 647 (goat polyclonal) | Molecular Probes | Cat# A-21244, RRID: AB_2535812 | 1:400 |
| Antibody | Anti-rabbit Alexa 488 (goat polyclonal) | Molecular Probes | Cat# A-11008, RRID: AB_143165 | 1:400 |
| Antibody | Anti-mouse Alexa 647 (goat polyclonal) | Molecular Probes | Cat# A-21235, RRID: AB_2535804 | 1:400 |
| Antibody | Anti-guinea pig Alexa 488 (goat polyclonal) | Molecular Probes | Cat# A-11073, RRID: AB_2534117 | 1:400 |
| Antibody | Anti-chicken Alexa 488 (goat polyclonal) | Molecular Probes | Cat# A-11039, RRID: AB_2534096 | 1:400 |
| Antibody | Anti-rat Alexa 647 (goat polyclonal) | Molecular Probes | Cat# A-21247, RRID: AB_141778 | 1:400 |
| Antibody | Anti-guinea pig Alexa 647 (goat polyclonal) | Molecular Probes | Cat# A-21450, RRID: AB_2535867 | 1:400 |
| Sequence-based reagent | Antisense probe pairs for in situ hybridisation chain reaction (HCR) | This study | DNA Oligos | See Supporting Zip Document 1 |
| Software, algorithm | RStudio, | RStudio | R version 4.0.3 | |
| Other | RDS | PMID:33125872 | NCBI GEO: GSE156455 | 0 hr after puparium formation (APF) and 15 hr APF |
| Other | RDS | PMID:33149298 | NCBI GEO GSE142789 | 12 hr APF and 24 hr APF |
| Other | RDS | PMID:35388222 | NCBI GEO: GSE167266 | |
| Software, algorithm | GraphPad Prism 9 | GraphPad Prism 9 | GraphPad Prism version 9.4.1 | |
| Software, algorithm | Adobe Photoshop | Adobe Photoshop | Adobe Photoshop 2021 | |
| Software, algorithm | Adobe Illustrator | Adobe Illustrator | Adobe Illustrator 2021 | |
| Software, algorithm | Fiji-ImageJ | PMID:22743772 | | |
| Chemical compound, drug | HCR amplification buffer | Molecular Instruments | BAM02224 | |
| Chemical compound, drug | HCR wash buffer | Molecular Instruments | BPW02124 | |
| Chemical compound, drug | HCR hybridisation buffer | Molecular Instruments | BPH02224 | |

*Continued on next page*

Continued

| Reagent type (species) or resource | Designation | Source or reference | Identifiers | Additional information |
|---|---|---|---|---|
| Chemical compound, drug | HCR amplifier B3-H1-546 | Molecular Instruments | S030724 | |
| Chemical compound, drug | HCR amplifier B3-H2-546 | Molecular Instruments | S031024 | |
| Chemical compound, drug | HCR amplifier B3-H1-647 | Molecular Instruments | S040124 | |
| Chemical compound, drug | HCR amplifier B3-H2-647 | Molecular Instruments | S040224 | |
| Chemical compound, drug | Paraformaldehyde | Fisher Scientific | 28908 | 4% solution |
| Chemical compound, drug | DAPI | Sigma | D9542-1MG | (1 µg/mL) |

### *Drosophila* stocks and maintenance

*D. melanogaster* strains and crosses were reared on a standard cornmeal medium and raised at 25 or 29°C or shifted from 18 to 29°C for genotypes as indicated in *Supplementary file 2*.

We used the following mutant and transgenic flies in combination or recombined in this study (see *Supplementary file 2* for more details; {} enclose individual genotypes, separated by commas):

Canton S, {ptc-lacZ/TM6B} (a gift from D Kalderon), {ywhsflp[122], tub-gal4, UAS-GFP; FRT40A, Gal80/Cyo} (a gift from F Schweisguth), {ey-Gal80; sp/Cyo;} (BDSC: 35822), {;Gal80ts; TM2/TM6B} (BDSC: 7108), {w[1118];; R27G05-Gal4} (BDSC: 48073), {;;UAS-CD8::GFP} (BDSC: 32187), {y[1] sc[*] v[1] sev[21]; Ci-RNAi} (BDSC: 64928), {UAS-Ci[76]} (a gift from J Treisman), {y[1]; FRT42D, ptc[S2]/CyO} (BDSC: 6332), {ywhsflp[122], tub-gal4, UAS-RFP; FRT42D, Gal80/Cyo} (a gift from G Struhl), {FRT40A, smo[3] /CyO} (a gift from C Boekel), {yw[1118]; hh::sfGFP} (BDSC: 86271), {w[1118]; hh[ts2] e[s]/TM6B} (BDSC: 1684), {;;ry[506], hh[P30]} (BL5530), {w[*]; GMR-Gal4;} (BL1104), {;;UAS-hh-RNAi} (BL32489).

### Mosaic analysis

We generated smo[3] and ptc[S2] MARCM (*Figure 3—figure supplement 1A* and *Figure 3I and J*, respectively) clones by heat-shocking larvae 2 days after egg laying (AEL) at 37°C for 90 min. To generate one wild-type MARCM clone per lamina (*Figure 1—figure supplement 1B, C*), we heat-shocked larvae (1 day AEL) for 60 min at 37°C. All MARCM crosses were raised at 25°C until dissection at 0–5 hr APF.

### Immunocytochemistry, antibodies, and microscopy

We dissected eye-optic lobe complexes from early pupae (0–5 hr APF) in 1× phosphate-buffered saline (PBS), fixed in 4% formaldehyde for 20 min, blocked in 5% normal donkey serum, and incubated in primary antibodies diluted in the block for two nights at 4°C. Samples were then washed in 1× PBS with 0.5% TritonX (PBSTx), incubated in secondary antibodies diluted in the block, washed in PBSTx, and mounted in SlowFade (Life Technologies).

We used the following primary antibodies in this study: mouse anti-Dac[2-3] (1:20, Developmental Studies Hybridoma Bank; DSHB), rat anti-Elav (1:100, DSHB), mouse anti-Elav (1:20, DSHB), mouse anti-Svp (1:50, DSHB), rabbit anti-Slp2 (1:100; a gift from C Desplan), rabbit-Bsh (1:500; a gift from C Desplan), mouse anti β-Gal (1:500; Promega #Z3781), chicken anti-GFP (1:400; EMD Millipore), mouse anti-Sim (1:20; a gift from T Tabata), chicken anti-RFP (1:500; Rockland #600-901-379s), rabbit-anti-GFP (1:500; Thermofisher #A6455), AlexaFluor405 conjugated goat anti-HRP (1:100; Jackson Immunolabs), AlexaFluor405-, Cy3-, or AlexaFluor647-conjugated goat anti-HRP (1:200; Jackson Immunolabs). Secondary antibodies were obtained from Jackson Immunolabs or Invitrogen and used at 1:800. Images were acquired using Zeiss 800 and 880 confocal microscopes with 40× objectives.

### In situ HCR

To assess putative lamina neuron-type-specific marker genes in vivo, we designed HCR probes against *slp2, svp, bsh, rdo, erm, VGlut*, and *dpr8*. We designed 6–21 antisense probe pairs against each target gene, tiled along the annotated transcripts but excluding regions of strong sequence similarity to other transcripts, with the corresponding initiator sequences for amplifiers B3 and B5 (*Choi et al., 2018*). We purchased HCR probes as DNA oligos from Thermo Fisher (at 100 µM in water and frozen). All probe sequences are included as source data (see Supporting Zip Document 1).

Eye-optic lobe complexes were dissected, fixed, and permeabilised as above. Samples were incubated in probe hybridisation buffer at 37°C for 30 min before being incubated with probes at 0.01 µM at 37°C overnight. Samples were washed four times for 15 min at 37°C with probe wash buffer and then two times for 5 min with 5× saline-sodium citrate with 0.001% Tween 20 solution (to make a 20× SSCT solution in distilled $H_2O$, 58.44 g/mol sodium chloride, 294.10 g/mol sodium citrate, pH adjusted to 7 with 14 N hydrochloric acid, with 0.001% Tween 20). Samples were incubated in an amplification buffer for 10 min. Hairpins H1 and H2 for each probe were snap-cooled (hairpins were heated to 95°C for 90 s and cooled to room temperature for 30 min) separately to avoid hairpin oligomerisation. About 12 pmol of each hairpin was added to samples in an amplification buffer and incubated overnight at room temperature. Samples were washed for 10 min in SSCT and then incubated in darkness at room temperature with 1:15 dilution of DAPI (Sigma: D9542) for 90 min. Samples were washed in 1× PBS for 30 min and mounted as above.

## scRNAseq analyses

To maximise temporal resolution during development as well as the number of cells analysed, we combined three publicly available scRNAseq datasets of optic lobes from the following developmental timepoints: wandering third instar larva, 0-hr APF, 12-hr APF, 15-hr APF, and 24-hr APF (NCBI GEO: GSE156455, GSE167266, GSE142789) (*Konstantinides et al., 2022*; *Kurmangaliyev et al., 2020*; *Özel et al., 2021*). We combined these datasets using the Seurat v.3 integration pipeline to remove batch effects between libraries (*Stuart et al., 2019*) as follows: using the default parameters in Seurat v4.0.1 we first normalised each dataset with the NormaliseData function. Next, we extracted the 2000 most variable features with the FindVariableFeatures function. We then integrated the data using the FindIntegrationAnchors and IntegrateData functions. Next, we clustered the integrated dataset using the following functions: ScaleData, RunPCA (using 150 principal components as in *Konstantinides et al., 2021*), FindNeighbours (80 dimensions), FindClusters (resolution = 5), and RunUMAP.

We annotated clusters corresponding to lamina cell types based on a combination of previous annotations from the source datasets (*Özel et al., 2021*) and known markers: *dac, eyes absent (eya), tailless (tll), glial cells missing (gcm)* for LPCs; *svp* and *slp2* for L1s; *slp2* for L2s; *erm* and *slp2 for L3s; bsh and ap for L4s; bsh* and *slp2 for L5s* (*Chotard et al., 2005*; *Guillermin et al., 2015*; *Hasegawa et al., 2013*; *Huang and Kunes, 1996*; *Piñeiro et al., 2014*; *Tan et al., 2015*).

To identify additional lamina-neuron-type markers, we used the FindMarkers (two-sided Wilcox rank-sum test) function with default parameters to identify positively expressed genes based on log-fold change in individual lamina neuron-type clusters relative to the other four lamina neuron clusters.

To analyse differentially expressed genes between the L1–L4 and the L5 convergent stream, we first used the CellSelector function to manually select the two streams as two individual clusters. Next, we used the FindMarkers (two-sided Wilcox rank-sum test) function with default parameters to identify positively or negatively expressed genes based on log-fold change. To visualise UMAPs and gene expression, we used the DimPlot and FeaturePlot functions.

## Quantification and statistical analyses

We used Fiji-ImageJ (*Schindelin et al., 2012*) or Imaris (version x64-9.5.1) to process and quantify confocal images as described below. We used Adobe Photoshop and Adobe Illustrator software to prepare figures. We used GraphPad Prism8, JMP, and R (version 4.0.3) software to perform statistical tests. In all graphs, whiskers indicate the SD. N values are indicated on graphs.

## MFI quantifications and statistical analyses
### Ptc-lacZ
Only optic lobes oriented with laminas in perfect cross-section were used for these quantifications. In Fiji-ImageJ, using photoreceptor axons (HRP), and the lobula plug (Dac expression) as landmarks, we selected the 10 most centrally located focal slices of the lamina (*ptc-lacZ*; *Figure 1I*; step size = 1mm) and generated average intensity projections of these. To generate MFI profile plots, we drew a line across each row from the youngest lamina column to the oldest column for each of the 6 rows (distal-proximal cell positions) of the lamina and measured β-Gal MFI. We then calculated the average MFI profile plot per row and generated line graphs to represent this data. We used a mixed effects linear model in JMP to test for an interaction between *ptc-lacZ (β-Gal)* MFI, distal-to-proximal cell position

and along the anterior-posterior axis starting from the first column (Summary Statistics are provided in *Supplementary file 1*). We used GraphPad Prism8 to apply a moving average of 6 neighbours to smooth the data, which are plotted in *Figure 1J*.

### hh-sfGFP/+

We cultured brain explants dissected from 0-hr APF *hh-sfGFP/+* animals (*Bostock et al., 2020*). As above, we obtained 10 centrally located optical slices and measured the MFI profiles along the distal-proximal positions of the youngest lamina columns (*Figure 2B and C*). The distal most lamina position was taken as the starting position, with the most proximal lamina precursor as the final. We normalised each MFI profile to the maximum MFI value for each profile. We then performed regression averaging using *ggplot* in R (version 4.0.3) to generate a plot of mean Hh::GFP fluorescence based on the individual profiles.

## Cell-type quantifications

### Slp2, Svp, and Bsh

We were unable to distinguish between L2 and L3 neurons since both express Slp2 only; therefore, we quantified the number of L2-L3s, L1s, L4s, and L5s in 10 centrally located optical slices in 10 optic lobes. Since total lamina neuron numbers were also affected by Hh pathway manipulations, we calculated the proportion of each lamina neuron type relative to the total number of neurons and plotted these normalised to the control in *Figures 3 and 4*. *Figure 3—figure supplement 1* shows the raw numbers of each lamina neuron type in each focal slice for each condition.

### *rdo, erm, VGlut, dpr8,* and *bsh*

In Fiji-ImageJ, we measured the area of the lamina (relative to the whole lamina) that was positive for *rdo* expression (but negative for *erm*) for L2 neurons; *erm* expression for L3 neurons; *VGlut* for L1 neurons; *ap* for L4 neurons, *bsh* and *dpr8* co-expression for L5 neurons per optical slice for the 10 most centrally located slices in 10 optic lobes. We then calculate the proportion relative to control for each of these markers and plotted these Graphpad Prism.

## Acknowledgements

We thank C Desplan, D Kalderon, B Shilo, G Struhl, and J Treisman for reagents, and S Ackerman, M Amoyel, B Conradt, C Desplan, A Franz, P Salinas, A Rossi, C Stern, L Venkatasubramanian and members of the Fernandes lab for comments on the manuscript. Stocks obtained from the Bloomington *Drosophila* Stock Center (NIH P40OD018537) were used in this study. Monoclonal antibodies obtained from the Developmental Studies Hybridoma Bank, created by the NICHD of the NIH and maintained at The University of Iowa, were used in this study. Wellcome Trust Sir Henry Dale Research Fellowship to VMF (210472/Z/18/Z), UCL Biosciences Graduate Research Scholarship to MPB, UCL Overseas Research Scholarship and UCL Graduate Research Scholarship to ARP, and UCL Research Opportunities Scholarship to AD.

## Additional information

### Competing interests

Vilaiwan M Fernandes: Reviewing editor, *eLife*. The other authors declare that no competing interests exist.

### Funding

| Funder | Grant reference number | Author |
| --- | --- | --- |
| Wellcome Trust | 210472/Z/18/Z | Vilaiwan M Fernandes |
| University College London | Biosciences Graduate Research Scholarship | Matthew P Bostock |

| Funder | Grant reference number | Author |
|---|---|---|
| University College London | Overseas Research Scholarship and Graduate Research Scholarship | Anadika R Prasad |
| University College London | Research Opportunity Scholarship | Alicia Donoghue |

The funders had no role in study design, data collection and interpretation, or the decision to submit the work for publication. For the purpose of Open Access, the authors have applied a CC BY public copyright license to any Author Accepted Manuscript version arising from this submission.

## Author contributions

Matthew P Bostock, Conceptualization, Formal analysis, Validation, Investigation, Visualization, Methodology, Writing - review and editing; Anadika R Prasad, Validation, Investigation, Visualization, Methodology, Writing - review and editing; Alicia Donoghue, Investigation; Vilaiwan M Fernandes, Conceptualization, Supervision, Funding acquisition, Investigation, Visualization, Methodology, Writing - original draft, Project administration, Writing - review and editing

## Author ORCIDs

Matthew P Bostock  http://orcid.org/0000-0003-1314-8998
Vilaiwan M Fernandes  http://orcid.org/0000-0002-1991-7252

## Decision letter and Author response

Decision letter https://doi.org/10.7554/eLife.78093.sa1
Author response https://doi.org/10.7554/eLife.78093.sa2

# Additional files

## Supplementary files

• Supplementary file 1. Table showing summary statistics of a mixed effects linear model for *ptc-lacZ* (*β-Gal*) MFI as a function of distance and cell position (distal-proximal).

• Supplementary file 2. Table listing all genotypes and experimental conditions used by figure panel. (Note that only female genotypes are listed though both sexes were included in our analyses).

• Supplementary file 3. Excel file containing all the probe sequences used for in-situ hybridisation chain reaction in this study.

• Transparent reporting form

## Data availability

All data generated or analysed during this study are included in the manuscript.

The following previously published datasets were used:

| Author(s) | Year | Dataset title | Dataset URL | Database and Identifier |
|---|---|---|---|---|
| Simon F, Jafari S, Holguera I, Chen Y-C, Benhra N, Naja El-Danaf R, Kapuralin K, Amy Malin J, Konstantinides N, Desplan C, NesetÖzel M | 2021 | Neuronal diversity and convergence in a visual system developmental atlas | https://www.ncbi.nlm.nih.gov/geo/query/acc.cgi?acc=GSE142789 | NCBI Gene Expression Omnibus, GSE142789 |
| Kurmangaliyev YZ, Yoo J, Valdes-Aleman J, Piero S, S Lawrence Z | 2020 | Transcriptional programs of circuit assembly in the *Drosophila* visual system | https://www.ncbi.nlm.nih.gov/geo/query/acc.cgi?acc=GSE156455 | NCBI Gene Expression Omnibus, GSE156455 |

*Continued on next page*

*Continued*

| Author(s) | Year | Dataset title | Dataset URL | Database and Identifier |
|---|---|---|---|---|
| Konstantinides N, Rossi AM, Escobar A, Dudragne L, Chen Y-C, Tran T, Jaimes AM, NesetÖzel M, Simon F, Shao Z, Tsankova NM, Fullard JF, Walldorf U, Roussos P, Desplan C | 2021 | A comprehensive series of temporal transcription factors in the fly visual system | https://www.ncbi.nlm.nih.gov/geo/query/acc.cgi?acc=GSE167266 | NCBI Gene Expression Omnibus, GSE167266 |

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
