## [Editor Report]

This manuscript uncovers a role for a Hh gradient in the differentiation of neuron types in the lamina of the fly eye, a phenomenon reminiscent of its role in the vertebrate nervous systems. It will be of special interest to those who study optic lobe development, but also of more general interest to developmental neurobiology.

---

## [Decision Letter]

**Decision letter after peer review:**

Thank you for submitting your article "Photoreceptors generate neuronal diversity in their target field through a Hedgehog morphogen gradient in *Drosophila*" for consideration by *eLife*. Your article has been reviewed by 3 peer reviewers, including Sonia Sen as Reviewing Editor and Reviewer #1, and the evaluation has been overseen by K VijayRaghavan as the Senior Editor.

Essential revisions:

We had three major concerns that we'd request the authors to address:

1. The gradient of Ptc-LacZ – The gradient wasn't obvious to us and we're not sure how it was quantified figure 1J. Could you please explain the quantification in figure 1J better? While revisiting the text around this it would be nice if the authors highlighted the two axes in which the gradient of ptc-LacZ plays out – proximal to distal and young to old.

2. The gradient of hh:

a. Is hh produced by the LPCs or is it a consequence of PR derived hh? Could you please test this by removing hh from the PRs and measuring it in the lamina?

b. The gradient itself wasn't very convincing. Figure 2C shows a Hh-GFP gradient in the distal lamina, but it looks essentially flat in the proximal half. Is it possible to demonstrate the gradient better? Some suggestions are – by performing IHC without permeabilization (while dissecting on ice); use an antibody against hh (instead of, or in addition to, hh:GFP).

c. Consequence of this gradient: The intermediate hh identities haven't been addressed sufficiently. Could the authors support their temperature sensitive allele data with others that modulate hh (or its activity) mildly? For example, by using clones homozygous for weak alleles of signalling pathway components, or expressing RNAi with a weaker GAL4 driver or at a lower temperatures. While doing so, could they use additional cell fate markers for these cell types?

3. Developmental origin: The data presented here are not sufficient to make any claims about developmental origin of the lamina neurons. To address it thoroughly would require a substantial set of new experiments (see detailed review). We felt that the hh gradient story will be of interest even without this data, and so recommend that the authors omit it from this manuscript. If they would like to include it, we recommend using MARCM or Dual MARCM with smaller clone sizes.

*Reviewer #1 (Recommendations for the authors):*

– Early developmental events in the lamina: I am not sure I understand the events that constitute early neurogenesis in the lamina. The authors talk about induction of the lamina precursors that then divide terminally. Are LPC similar to neuroblasts? Or are they more similar to GMCs, which divides terminally? Could the authors address what's known about these early events in more detail the introduction?

– Shared developmental origin of the L1-L4 and L5: Since the authors integrate scRNAseq data across developmental time, I wonder if the trajectories they see are not in fact temporal differences in specification, rather than different developmental origins? Given the size of the MARCM clone used to test this, I'm not sure that any claim about developmental origin can be made. The clone looks rather like a developmental compartment (or a spatial domain in the lamina). To really address developmental origin of the neurons, it would have been nice to have analysed multiple small clones and determine whether all 5 neuron types are always seen together.

– The gradients: Both the Hh and the ptc gradients follow a similar profile – high distally, low proximally. In the absence of Hh, ptc acts as a negative regulator of Hh, but in its presence, it activates it. Would this mean that effective Hh signalling will be more or less similar across the LCPs? (Although, the gradients of ptc and sim were not obvious to me.)

*Reviewer #2 (Recommendations for the authors):*

1) The claim that a gradient of Hh is meaningful, rather than simply its presence or absence, needs to be better supported by finding a way to produce intermediate levels of Hh signaling that significantly increase the numbers of L1 and/or L4 neurons. No increase is seen in Figure 3, and the differences are marked as "ns" in Figure 4. This should also be done in a way that does not affect the timing of exposure to Hh. Some possible methods would be by using clones homozygous for weak alleles of signaling pathway components, or expressing RNAi with a weaker GAL4 driver or at a lower temperature.

2) The authors should provide more than a single marker for cell fates, and ideally would also distinguish between L2 and L3 fates. For instance, different lamina neurons can be distinguished by the expression of specific GAL4 or LexA drivers, or by the medulla layers in which they arborize.

*Reviewer #3 (Recommendations for the authors):*

This paper is well-written, easy to follow, and the data mostly appear to be convincing in support of the key claims. There are a few experiments that I think could potentially help to elucidate the origin of Hh.

1) As the authors mentioned, it is hard to tell whether the Hh distribution is from secreted or intracellular Hh. Secreted proteins can be harder to localize after dissection and staining protocols than intracellular proteins, and therefore the images and quantifications of dissected laminas could be missing some information. In order to have a better sense of the distribution in vivo, it could be informative to use in vivo multiphoton imaging of the developing visual system (such as in PMID: 24444078) with the Hh::GFP animals. Alternatively, you could also use IHC and stain without permeabilization, which should reduce antibody access to intracellular Hh::GFP. To distinguish from endogenous GFP, a secondary antibody with a separate fluorophore could be used.

2) To identify photoreceptors as the definitive source of the Hh signaling, it would be helpful to reduce Hh signaling specifically in photoreceptors and see whether L5 neurons are still specified at higher levels.

---

## [Author Response]

Essential revisions:We had three major concerns that we'd request the authors to address:1. The gradient of Ptc-LacZ – The gradient wasn't obvious to us and we're not sure how it was quantified figure 1J. Could you please explain the quantification in figure 1J better? While revisiting the text around this it would be nice if the authors highlighted the two axes in which the gradient of ptc-LacZ plays out – proximal to distal and young to old.

We have updated the text to clarify how we measured the levels of *ptc-LacZ* along both anterior-posterior (*i.e.* young to old) (Figure 1J) and proximal to distal axes (new panel: Figure 1K) in the main text and in the Materials and methods section. We have also highlighted the two relevant axes, young to old and distal to proximal, in Figure 1I”’.

The revised main text now reads (Page 4, Lines 17-37):

“To validate these data in vivo we examined expression of a transcriptional reporter for *ptc (ptc-lacZ)* as a readout of Hh signalling activity (Chen and Struhl, 1996; Tabata and Kornberg, 1994) (Figure 1I)*.* We measured the mean fluorescence intensity of b-Galactosidase (b -Gal) in the lamina at six cell positions corresponding to each LPC position within a column (*i.e.,* position 1 corresponded to the most distal cell (prospective L2) and position 6 corresponded to the most proximal cell (prospective L5) in columns; schematic in Figure 1J) from young to old columns (anterior to posterior axis). To do so we generated average intensity projections (from 10 optical slices) obtained from the central lamina and then measured the fluorescence intensity profiles for each of the 6 cell positions from the youngest column to the oldest column (quantified in Figure 1J with summary statistics from a mixed effects linear model in Table S1; see Materials and methods for more detail). We also measured the mean fluorescence intensity of b -Gal for each cell along the distal to proximal axis for the youngest column (Figure 1K). Consistent with the transcriptomic data where *ptc* expression was lowest in older maturing neurons (Figure 1E), *ptc-lacZ* expression eventually decreased along the young to old axis for each of the six (distal-proximal) cell positions (Figure 1I,J; N=10). Importantly, in young columns prior to neuronal differentiation, we observed a gradient of *ptc-lacZ* expression, which was highest in the distal lamina and decreased along the distal-proximal axis (Figure 1K; one-way ANOVA with Dunn’s multiple comparisons test, P<0.0001). Thus, we observed that the direct Hh signalling target, *ptc*, was graded along the distal to proximal length of the youngest column, indicating that there is a gradient of Hh signalling activity in lamina columns prior to neuronal differentiation.”

2. The gradient of hh:a. Is hh produced by the LPCs or is it a consequence of PR derived hh? Could you please test this by removing hh from the PRs and measuring it in the lamina?

To clarify, we have understood (from the individual reviewer comments that follow) that the reviewers would like us to confirm that it is Hh from photoreceptors that forms a gradient and specifies lamina neuron identities. To do so, we should remove Hh from photoreceptors to see a matching effect on lamina patterning, rather than on the Hh gradient itself.

In the eye disc, Hh promotes morphogenetic furrow progression and further photoreceptor development. Therefore, removing it from photoreceptors completely precludes analysis in the lamina as the lamina does not develop in the absence of photoreceptors (Huang and Kunes, 1996).

Instead, to address this question, we first examined *hh-lacZ* (*hh^P30^*) expression in eye-disc-optic lobe complexes (Figure 6A) and found that it was only expressed in photoreceptors and a few rare cells near the central brain. This is consistent with findings from Huang and Kunes (1996), who previously reported that only photoreceptors expressed Hh by immunohistochemistry (IHC) (Huang and Kunes, 1996; Figure 3).

We also used GMR-Gal4 to misexpress a previously validated RNAi against Hh to reduce Hh levels in photoreceptors partially. Photoreceptor development and early lamina development proceeded normally, indicative of a very mild Hh knockdown in photoreceptors. Nonetheless, this manipulation resulted in significant patterning defects in the lamina such that there was an increase in the proportion of Svp and Slp2 expressing L1 neurons (an intermediate cell identity). Importantly, L1 neurons were no longer restricted to a single row at intermediate positions in lamina columns but instead were observed at more distal positions. Thus, mildly decreasing Hh expression in photoreceptor caused lamina patterning defects consistent with generating more intermediate Hh signalling conditions (see below).

We have added an additional figure with these data – Figure 6 – and included a description of these data in the results and Discussion section.

The revised manuscript now reads (Page 8, Lines 4-28):

“Photoreceptor-derived Hh specifies lamina neuron identities

Others have reported previously that photoreceptors are the sole source of Hh to the optic lobes during lamina development (Huang and Kunes, 1996). We corroborated these results using an enhancer trap in the *hh* locus *(hh-lacZ; hh^P30^),* which showed that b-Gal expression was restricted to photoreceptors in the eye disc and absent from the optic lobes, except for a few sparse cells near the central brain (Figure 6A). We reasoned that if photoreceptor-derived Hh indeed patterns lamina neuron identities, then disrupting Hh expression specifically in photoreceptors should in turn result in lamina neuron patterning defects. Hh secreted by developing photoreceptors is required to drive morphogenetic furrow progression and continued photoreceptor development in the eye disc (Greenwood and Struhl, 1999; Treisman, 2013). Consequently, removing *hh* expression completely in the eye disc has been shown to block both photoreceptor development (Greenwood and Struhl, 1999; Treisman, 2013) and lamina development, due to its early role in inducing LPCs (Huang and Kunes, 1998, 1996). Therefore, we used *GMR-Gal4*, which is expressed after photoreceptor birth together with a previously validated RNAi line (Kim et al., 2014; Ni et al., 2011; Sahai-hernandez and Nystul, 2013) to partially knock down Hh in photoreceptors, thus bypassing its early roles. Under these conditions, lamina size was not noticeably affected, indicative of a mild knockdown (Figure 6B,C). When we evaluated lamina neuron type identities, we found that Slp2 and Svp expressing cells (L1s) increased by 1.2 fold relative to controls (Figure 6C; Mann-Whitney U test; P<0.001; N=10), consistent with more lamina precursor cells experiencing intermediate Hh signalling activity. Importantly, L1 neurons were no longer restricted to a single row at intermediate positions in columns but instead were recovered in more distal positions (Figure 6C). Thus, partially knocking down Hh expression in photoreceptors disrupted lamina neuron patterning in favour of more intermediate cell identities (L1s), consistent with mildly decreasing Hh signalling to more intermediate levels in the lamina.”

b. The gradient itself wasn't very convincing. Figure 2C shows a Hh-GFP gradient in the distal lamina, but it looks essentially flat in the proximal half. Is it possible to demonstrate the gradient better? Some suggestions are – by performing IHC without permeabilization (while dissecting on ice); use an antibody against hh (instead of, or in addition to, hh:GFP).

On the suggestion of Reviewer #3 we opted to use 2-photon live-imaging on cultured brain explants to visualise Hh::GFP without need for fixation and IHC. Indeed, while visualizing the gradient using IHC showed a relatively flat gradient in the proximal half of the lamina, imaging Hh::GFP in live samples showed a much more gradual decline in Hh::GFP mean fluorescence intensity. We thank the reviewers for this suggestion as it has significantly strengthened our data. We have added these images and quantifications to Figure 2B,C and amended the main text as follows on page 5, lines 6-16:

“We first used immunohistochemistry (IHC) to detect Hh::GFP and observed higher levels in younger photoreceptor cell bodies compared to older photoreceptor cell bodies in the eye disc, consistent with previous reports (Figure 2A) (Huang and Kunes, 1996). We also observed Hh::GFP in photoreceptor axons in the optic stalk, but observed very little signal in photoreceptor axons in the lamina (Figure 2A; N=18). Secreted proteins can be particularly sensitive to fixation and washes during IHC, therefore to avoid any confounding artefacts potentially caused by IHC, we visualised Hh::GFP live in cultured brain explants (Bostock et al., 2020) (Figure 2B). We measured Hh::GFP mean fluorescence intensity as a function of distal-proximal distance for the youngest lamina column (see Materials and methods) and observed a gradient polarised from high to low from the distal to proximal ends of columns (Figure 2B-C; N=10).”

c. Consequence of this gradient: The intermediate hh identities haven't been addressed sufficiently. Could the authors support their temperature sensitive allele data with others that modulate hh (or its activity) mildly? For example, by using clones homozygous for weak alleles of signalling pathway components, or expressing RNAi with a weaker GAL4 driver or at a lower temperatures. While doing so, could they use additional cell fate markers for these cell types?

Firstly, we apologise for an error in our original manuscript in Figure 3 (from our original submission; now Figure 5), which indicated that the increase in the proportion of L1 neurons (Svp and Slp2 coexpressing cells) was not significant for *hh^ts2^* animals when shifted to the restrictive temperature for 6 hours. Indeed, we did observe a significant increase in L1s (One way ANOVA with Dunn’s multiple comparisons test; P<0.05; N=9). We have now corrected this mistake in the figure (Now Figure 5).

Nonetheless, we agree with the reviewers and have provided additional data to support the claim that intermediate Hh signalling levels are instructive of intermediate cell fates.

First, in response to the point made by Reviewer #2, we identified additional lamina neuron type-specific markers (see below) using published single cell RNA sequencing data (Konstantinides et al., 2022; Kurmangaliyev et al., 2020; Özel et al., 2021) and validated marker gene expression in vivo using in situ hybridisation chain reaction (Choi et al., 2018, 2016). We thank the reviewer for this suggestion as it has increased our confidence in identifying the specific lamina cell types and strengthened our conclusions. These data are found in Figure 1—figure supplement 1B-K, and described in the main text on pages 3 and 4, lines 38-40 and 1-8, respectively:

“First, we probed these data to identify additional early markers of cell identity for each of the lamina neuron types (beyond Slp2, Bsh and Svp). Consistent with previous reports, scRNAseq analysis showed that L3 and L4 neurons expressed *earmuff (erm)* and *apterous (ap)*, respectively (Tan et al., 2015; Ting et al., 2005). In addition, we found that *reduced ocelli (rdo)* was expressed at high levels in L2 neurons and at lower levels and more sporadically in L3 neurons, that L1 neurons expressed *Vesicular Glutamate transporter* (*VGlut) specifically,* and that L5 and L3 neurons expressed high and low levels of *defective proboscis extension response 8 (dpr8)*, respectively (Figure1—figure supplement 1B-F). We used in situ hybridisation chain reaction (HCR) (Choi et al., 2018) to validate marker expression in vivo at 0 hours after puparium formation (0h APF) (Figure 1—figure supplement 1G-K), thus providing additional markers for the differentiating lamina neuron types.”

Next, to achieve intermediate levels of Hh signalling activity, we used a lamina-specific Gal4 (*R27G05-Gal4)* to express Ci^repressor^
*(lamina^ts^>Ci^rep^)* at two temperatures: 29°C (for a strong inhibition of Hh signalling) and 25°C (for a weaker inhibition of Hh signalling). Using multiple lamina neuron type specific markers, we observed that intermediate cell identities (L1s and L4s) increased when Hh signalling was weakly inhibited (*lamina^ts^>Ci^rep^* at 25°C), whereas proximal cell identities increased when Hh signalling was inhibited strongly (*lamina^ts^>Ci^rep^* at 29°C). These results have been added in as a new figure – Figure 4 together with Figure 4—figure supplement 1, and the main text has been amended on pages 6 and 7, lines 38-40 and 1-15, respectively:

“Therefore, we attempted to tune Hh signalling to intermediate levels by using the Gal4/UAS system to express Ci^rep^ in the lamina as before under two temperature conditions (in the presence of Gal80^ts^): 29°C for a strong inhibition of Hh signalling (as above) and 25°C for a milder inhibition. We then evaluated the distribution of lamina neuron types using *rdo, erm, VGlut, ap, dpr8* and *bsh.* Strong inhibition of Hh signalling (*lamina^ts^>Ci^rep^* at 29°C) led to a 2.2-fold increase relative to controls in the proportion of the lamina that co-expressed *dpr8* and *bsh* (*i.e.* L5s) at the expense of other markers (Figure 4A- F, J-N; N*^dpr8+bsh^*=8; one-way ANOVA with Dunn’s multiple comparisons test; P*^dpr8+bsh^*<0.0001). As before, *dpr8* and *bsh* coexpressing cells (L5s) were no longer restricted to a single row in the proximal lamina but were distributed along the entire distal-proximal lengths of columns (Figure 4C,F). In contrast, milder inhibition of Hh signalling (*lamina^ts^>Ci^rep^* at 25°C) led to a 2.5-fold increase relative to controls in the proportion of the lamina that expressed *VGlut* (*i.e.* L1s), and a 1.9-fold increase relative to controls in the proportion of the lamina that expressed *ap* (*i.e.* L4s), at the expense of *rdo* expressing *(erm-*negative*)* L2s (Figure 4A-C, G-I, J-N; one-way ANOVA with Dunn’s multiple comparisons test; P*^VGlut^*<0.0001; P*^ap^*<0.01). We obtained similar results when we evaluated the distribution of neuron types with Slp2, Svp and Bsh (Figure 4—figure supplement 1). Thus, intermediate neuron identities (L1s and L4s) were favoured when we tuned Hh signalling to intermediate levels.”

3. Developmental origin: The data presented here are not sufficient to make any claims about developmental origin of the lamina neurons. To address it thoroughly would require a substantial set of new experiments (see detailed review). We felt that the hh gradient story will be of interest even without this data, and so recommend that the authors omit it from this manuscript. If they would like to include it, we recommend using MARCM or Dual MARCM with smaller clone sizes.

We agree with the reviewers that these experiments are not central to our argument about Hh forming a morphogen gradient and as requested have removed these data from our manuscript.

Reviewer #1 (Recommendations for the authors):– Early developmental events in the lamina: I am not sure I understand the events that constitute early neurogenesis in the lamina. The authors talk about induction of the lamina precursors that then divide terminally. Are LPC similar to neuroblasts? Or are they more similar to GMCs, which divides terminally? Could the authors address what's known about these early events in more detail the introduction?

LPCs do not express classic neuroblast markers and thus represent a distinct type of progenitor. We are unaware of studies that have carefully evaluated whether mitotic LPCs resemble GMCs, however, generally, they are not thought to resemble GMCs as there is no obvious size difference between mitotic LPCs and their daughters. We have modified the text in our introduction to clarify this point as follows on page 2, line 26-31:

“LPCs do not express classic neuroblast markers but are instead a distinct progenitor type, which undergo terminal divisions in response to Hh signalling (Apitz and Salecker, 2014). Finally, Hh signalling also promotes LPC adhesion to photoreceptor axons, which facilitates their assembly into columns (*i.e.* ensembles of stacked LPC cell bodies associated with photoreceptor axon bundles; Figure 1A, B) (Huang and Kunes, 1998, 1996; Sugie et al., 2010; Umetsu et al., 2006).”

– Shared developmental origin of the L1-L4 and L5: Since the authors integrate scRNAseq data across developmental time, I wonder if the trajectories they see are not in fact temporal differences in specification, rather than different developmental origins? Given the size of the MARCM clone used to test this, I'm not sure that any claim about developmental origin can be made. The clone looks rather like a developmental compartment (or a spatial domain in the lamina). To really address developmental origin of the neurons, it would have been nice to have analysed multiple small clones and determine whether all 5 neuron types are always seen together.

Agreed. Please see response to (3) above.

– The gradients: Both the Hh and the ptc gradients follow a similar profile – high distally, low proximally. In the absence of Hh, ptc acts as a negative regulator of Hh, but in its presence, it activates it. Would this mean that effective Hh signalling will be more or less similar across the LCPs? (Although, the gradients of ptc and sim were not obvious to me.)

This is an interesting hypothesis. In our manuscript we focus on *ptc-lacZ* solely as a reporter of Hh signalling activity (Chen and Struhl, 1996; Tabata and Kornberg, 1994).

Ptc antagonizes Smo, but when Hh binds Ptc, this relieves Ptc inhibition of Smo. Thus, Ptc is a negative regular of Hh signalling in the presence and absence of Hh ligand. Ptc is also shown to limit the range of Hh by binding and sequestering it (Chen and Struhl, 1996).

While we have not examined the contribution of Ptc to gradient formation in this study, our manipulations of Hh signalling levels drove matching changes in lamina neuron identities, which are consistent with LPCs experiencing high levels of Hh signalling in the distal lamina and low levels in the proximal lamina.

Unfortunately, we were not able to improve our images to show a gradient of Sim expression as DSHB no longer produces antibody against Sim. We have therefore removed these data from the manuscript.

Reviewer #2 (Recommendations for the authors):1) The claim that a gradient of Hh is meaningful, rather than simply its presence or absence, needs to be better supported by finding a way to produce intermediate levels of Hh signaling that significantly increase the numbers of L1 and/or L4 neurons. No increase is seen in Figure 3, and the differences are marked as "ns" in Figure 4. This should also be done in a way that does not affect the timing of exposure to Hh. Some possible methods would be by using clones homozygous for weak alleles of signaling pathway components, or expressing RNAi with a weaker GAL4 driver or at a lower temperature.

See response to point (2) from the combined reviews above. We have conducted new experiments to address the role of Hh in specifying intermediate identities, and we believe that our conclusions are strengthened as a result. We thank the reviewer for their careful analysis of our work.

2) The authors should provide more than a single marker for cell fates, and ideally would also distinguish between L2 and L3 fates. For instance, different lamina neurons can be distinguished by the expression of specific GAL4 or LexA drivers, or by the medulla layers in which they arborize.

Thank you for this suggestion, we agree with the reviewer. Please see response to (2c) above. With the additional markers that we identified, we were able to distinguish between L2 and L3 fates. Addressing this point has substantially improved our manuscript.

Reviewer #3 (Recommendations for the authors):This paper is well-written, easy to follow, and the data mostly appear to be convincing in support of the key claims. There are a few experiments that I think could potentially help to elucidate the origin of Hh.1) As the authors mentioned, it is hard to tell whether the Hh distribution is from secreted or intracellular Hh. Secreted proteins can be harder to localize after dissection and staining protocols than intracellular proteins, and therefore the images and quantifications of dissected laminas could be missing some information. In order to have a better sense of the distribution in vivo, it could be informative to use in vivo multiphoton imaging of the developing visual system (such as in PMID: 24444078) with the Hh::GFP animals. Alternatively, you could also use IHC and stain without permeabilization, which should reduce antibody access to intracellular Hh::GFP. To distinguish from endogenous GFP, a secondary antibody with a separate fluorophore could be used.

Thank you for this suggestion. Please see response to (2b) above.

2) To identify photoreceptors as the definitive source of the Hh signaling, it would be helpful to reduce Hh signaling specifically in photoreceptors and see whether L5 neurons are still specified at higher levels.

Please see response to (2a) above.